JGP Journal of General Physiology

# AMPK-mediated HCN4 channel phosphorylation contributes to age-related intrinsic bradycardia

Luca M.G. Palloni[1] , Nicole Sarno[1] , Caterina Azzoni[1] , Nicol Furia[1] , Matteo E. Mangoni[2] , Alessandro Porro[1] , Teresa Neeman[3] , Andrea Saponaro[4] , Gerhard Thiel[1,5] , Anna Moroni[1] , and Dario DiFrancesco[1]

The regulation of the hyperpolarization-activated cyclic nucleotide–gated 4 (HCN4) channels in pacemaker myocytes is essential for maintaining physiological cardiac rhythm. HCN4 dysfunctional behavior is among the major factors contributing to sinus node disease, a primary cause of pacemaker implantation. Previous work has shown that AMP-activated protein kinase (AMPK) activation leads to sinus bradycardia, a process attributable to cardiac remodeling that involves a decrease in HCN4 membrane expression, but the mechanism underlying this event remains unclear. We show here that AMPK can act as a posttranslational effector by phosphorylating Ser1157 at the C terminus of HCN4, a modification associated with a decrease in HCN4 membrane expression contributing to altered electrophysiological properties of cardiac pacemaker cells. Furthermore, we provide evidence that AMPK is constitutively activated in aged, but not young, mice, correlating with an increased development of intrinsic bradycardia. These findings support the view that AMPK is a key player in cardiac rhythm regulation and provide new insights into the molecular mechanisms underlying age-related changes in cardiac rhythm regulation.

## Introduction

A key mechanism providing essential contribution to physiological generation and control of spontaneous pacemaker activity of sinoatrial node (SAN) cells is activation of the "funny" current ($I_f$) (Brown et al., 1979) taking place during the early fraction of the diastolic depolarization phase of the action potential. By activating at the termination of an action potential, $I_f$ slowly depolarizes the membrane voltage of pacemaker cells until the threshold for a new action potential firing is reached (DiFrancesco, 1993; DiFrancesco, 2010).

The molecular determinants of the funny current $I_f$, the hyperpolarization-activated cyclic nucleotide–gated 1–4 (HCN1–4) channels, belong to the superfamily of voltage-dependent K+ (Kv) and the cyclic nucleotide–gated (CNG) channels (Gauss et al., 1998; Ludwig et al., 1999; Santoro et al., 1998). Hcn genes are widely expressed in the heart and nervous system, with different patterns of expression (Robinson and Siegelbaum, 2003). In the SAN, $I_f$ is mediated predominantly by the HCN4 isotype (Herrmann et al., 2011; Ishii et al., 1999; Marionneau et al., 2005), with contributions from HCN1 and HCN2 (Herrmann et al., 2011; Moroni et al., 2001).

f/HCN channels are modulated by the second messenger cAMP, which shifts the voltage activation toward depolarized potentials and increases the amount of current at any given voltage (DiFrancesco and Tortora, 1991; DiFrancesco and Tromba, 1988; Porro et al., 2020). This mechanism is critical in underlying the autonomic regulation of heart rate (HR) by the autonomic nervous system, which acts by modulating the cAMP concentration in pacemaker cells (Barbuti and DiFrancesco, 2008; DiFrancesco, 1991; DiFrancesco, 1995; DiFrancesco, 2006; DiFrancesco, 2010).

In addition to the acute, rapid modulation of HCN channels by the autonomic nervous system, also long-term events involving remodeling of ion channels' contributions to SAN cell activity can affect the sinus rhythm. Together with acute, dysfunctional regulation of HCN channels, also long-term effects can thus potentially create a substrate for arrhythmogenesis. These events, occurring gradually over time, affect the HR independently of the autonomous nervous system, and cause in elderly subjects an intrinsic bradycardia (Jose and Collison, 1970; Opthof, 2000; Ostchega et al., 2011). This occurs even though intrinsic bradycardia—defined as the slowing of pacemaker HR measured during autonomic blockade—can be compensated by changes in autonomic drive, such as a decrease in parasympathetic tone or an increase in sympathetic tone, thereby maintaining a relatively stable resting HR in older adults (Peters et al., 2020; Piantoni et al., 2021; Choi et al., 2021).

[1]Department of Biosciences, University of Milano, Milano, Italy;   [2]Institut de Génomique Fonctionnelle, Université de Montpellier, CNRS, INSERM, Montpellier, France;   [3]ANU College of Science, The Australian National University, Canberra, Australia;   [4]Department of Pharmacological and Biomolecular Sciences, University of Milano, Milano, Italy;   [5]Department of Biology, Membrane Biophysics and Centre for Synthetic Biology, TU-Darmstadt, Darmstadt, Germany.

Correspondence to Dario DiFrancesco: dario.difrancesco@unimi.it;   Luca Palloni: luca.palloni@unimi.it.

Although some of the mechanisms behind this pacemaker slowdown are being elucidated (Choi et al., 2021; Larson et al., 2013; Monfredi and Boyett, 2015), there are still many unanswered questions.

A potential causal link between aging and pacemaker slowdown can be found at the level of cardiac energy metabolism. The cardiac cellular homeostasis is directly linked to HR, which rises linearly in response to increases in myocardial oxygen consumption and the contractile state (Boerth et al., 1969). In the presence of excessive energy consumption and decreased ATP availability, the systemic energy-sensing AMP-activated protein kinase (AMPK) has been shown to activate and play a pivotal role in mediating cardiac remodeling, influencing the ion channel composition of pacemaker cells (Yavari et al., 2017). AMPK is a ubiquitously expressed serine/threonine kinase composed of a catalytic α subunit and regulatory β and γ subunits (Kim et al., 2016; Yan et al., 2018). It plays a key role in maintaining cellular energy homeostasis by promoting glucose uptake and fatty acid oxidation during periods of low energy availability (Hardie et al., 2012). In humans, activating mutations in the gene encoding the γ2 subunit of AMPK (*PRKAG2*) result in an autosomal dominant disorder whose heterogeneous phenotypic spectrum includes left ventricular hypertrophy and prominent electrophysiological disturbances (Blair et al., 2001; Gollob et al., 2001). Studies of transgenic mouse models suggest that constitutive activation of AMPK alters the intrinsic properties of pacemaker cells by downregulating several ion channels, including HCN4 (Yavari et al., 2017). While AMPK is known to affect cellular homeostasis through a variety of complex biochemical pathways (Hardie et al., 2012; Liu and Sabatini, 2020; Richter and Hargreaves, 2013), our results show that AMPK inhibits HCN4 channel function by a specific phosphorylation of Ser1157 at the C terminus and consequent reduction of membrane channel expression. Furthermore, our findings support the view that AMPK activation contributes to the intrinsic bradycardia commonly observed in elderly individuals.

# Materials and methods
## Animals
SAN pacemaker cells were isolated from C57BL/6J mice (IMSR_JAX:000664; Charles River Laboratories Italia S.r.l., RRID) according to the authorized license no. 839C7.N.2BF.

Gender-dependent selection criteria: In a first set of experiments aimed at identifying possible gender-dependent differences in response to drug treatment of SAN pacemaker cells, both male and female mice, aged 3 mo, were included in the study. Pregnant females were excluded. In a subsequent set of experiments, having verified a more robust response in male-derived cells, females were excluded from the study.

Age-related selection criteria: To investigate potential age-dependent changes in response to drug treatment, a specific set of experiments included only male mice aged either 3 mo (young) or 23–24 mo (old). Mice younger than 3 mo or between 4 and 22 mo were excluded from the study. The body weight of male mice included in the study ranged approximately from 25 to 30 g (young) and 50 to 55 g (old).

Mice were killed by cervical dislocation, and hearts were excised and immersed in normal Tyrode's solution (140 mM NaCl, 5.4 mM KCl, 1 mM $MgCl_2$, 1.8 mM $CaCl_2$, 5.5 mM D-glucose, and 5 mM Hepes, adjusted to pH 7.4 with NaOH) containing heparin preheated at 37°C. The SAN tissue was excised by cutting along the crista terminalis and the interatrial septum and transferred into a low-$Ca^{2+}$ solution containing 140 mM NaCl, 5.4 mM KCl, 0.5 mM $MgCl_2$, 0.2 mM $CaCl_2$, 1.2 mM $KH_2PO_4$, 50 mM taurine, 5.5 mM D-glucose, 1 mg/ml BSA, and 5 mM Hepes–NaOH (adjusted to pH 6.9 with NaOH). Enzymatic digestion was carried out for 25–30 min at 37°C in the low-$Ca^{2+}$ solution containing purified collagenase I and II (0.15 mg/ml Liberase medium Thermolysin, Roche) and elastase (0.5 mg/ml, Worthington). The digested tissue was washed and transferred to a modified "Kraftbrühe" solution containing 100 mM K-glutamate, 10 mM K-aspartate, 25 mM KCl, 10 mM $KH_2PO_4$, 2 mM $MgSO_4$, 20 mM taurine, 5 mM creatine, 0.5 mM EGTA, 20 mM D-glucose, 5 mM Hepes, and 1 mg/ml BSA (adjusted to pH 7.2 with KOH). Single cells were dissociated in Kraftbrühe solution at 37°C by manual agitation using a flame-forged Pasteur pipette. To recover the automaticity of the SAN cells, $Ca^{2+}$ was gradually reintroduced in the cell's storage solution to a final concentration of 1.8 mM. Cells were then dispersed on glass-bottom P35 dishes (Greiner Bio-One) coated with laminin (1–2 µg/ml, L2020; Merck Life Science S.r.l.).

## Cell culture
HEK293T cells were cultured in Dulbecco's modified Eagle's medium (Euroclone), and HEK293F in FreeStyle 293 Expression Medium (Gibco), both supplemented with 10% fetal bovine serum (Euroclone) and 1% penicillin–streptomycin (100 U/ml of penicillin and 100 µg/ml of streptomycin) (Euroclone). Cells were stored in a 37°C humidified incubator with 5% $CO_2$, and their maintenance was always carried out in sterile conditions under a laminar flow hood (Thermo Fisher Scientific, MSC-Advantage 1.2).

## Constructs
For patch clamp, the human *HCN4* gene was cloned in pcDNA3.1 (Invitrogen) (Addgene 100544; RRID), and mutations S1157A, S1157D, S1158A, S1158D, S1157A/S1158A, and S1157D/S1158D were introduced with the QuikChange II XL (Agilent Technologies) kit using 40- to 50-bp primers generated from QuikChange Primer Design Tool (Agilent Technologies). For FACS experiments, the hemagglutinin (HA)-tag was inserted into the extracellular S3–S4 loop using the Gibson assembly. The gene was then cloned into the pEGFP-C1 expression vector (Addgene 26674; RRID), positioning GFP at the N terminus of the channel. The cDNA encoding rabbit HCN4 (GenBank: NM_001082707) carrying an internal deletion (from residues 783 to 1064), hereafter HCN4ΔC, was cloned into a modified pEG BacMam vector (Goehring et al., 2014, Addgene 160451; RRID) (hereafter pEGA) for large-scale protein purification from mammalian cells. The internal deletion eliminates a poorly conserved region in the C-terminal portion of the HCN channel protein but preserves the extreme C terminus where Ser1129 (corresponding to Ser1157 in human) is located.

## Electrophysiology

Electrophysiology recordings of SAN and transfected HEK293 (CVCL 0045; RRID) cells were performed using an e-Patch amplifier (Elements) in the whole-cell configuration. SAN cells were perfused with Tyrode's solution containing 2 mM $BaCl_2$ and 2 mM $MnCl_2$, and heated at 37°C. Patch pipettes were fabricated from 1.5 mm O.D. and 0.86 I.D. HEK293T (CVCL 0063; RRID) and HEK293F (CVCL 6642; RRID) cells were kept in extracellular solution containing 110 mM NaCl, 30 mM KCl, 1.8 mM $CaCl_2$, 0.5 mM $MgCl_2$, 5 mM Hepes-KOH buffer, 5 mM $BaCl_2$, and 2 mM $MnCl_2$; pH 7.4 adjusted with NaOH. Borosilicate glass capillaries (Sutter) were pulled with P-97 Flaming/Brown Micropipette Puller (Sutter) and had resistances of 3–6 M$\Omega$. Pipettes were loaded with intracellular solution containing 80 mM KAsp, 50 mM KCl, 10 mM EGTA, 1 mM $MgCl_2$, 3 mM Na-ATP, and 10 mM Hepes-KOH; pH 7.2 adjusted with KOH.

For SAN cells, the voltage protocol consisted in 12 hyperpolarizing steps from –135 mV (0.6 s) to –25 mV (6.1 s), in 10-mV and 0.5-s increments from a holding potential of –35 mV. After each step, a 0.9-s pulse at –125 mV was applied to evaluate tail currents. A similar protocol was applied for HEK293 cells: eight hyperpolarizing steps from –130 mV (5 s) to –25 mV (26 s), in 15-mV and 2.5-s increments from a holding potential of –25 mV. Tail currents were measured with 2.5-s pulses at –135 mV. The IV relationship was generated by plotting the current at the steady state normalized to cell capacitance against voltage. The voltage activation curve was calculated by measuring tail currents and fitted according to the Boltzmann equation: $y(V) = 1/\{1 + \exp[(V – V_{1/2})/s]\}$, where $y(V)$ is the fractional current activation, $V_{1/2}$ is the half-activation voltage, and s is the inverse-slope factor.

The fully activated I/V relationship was measured according to a previously developed protocol in the voltage range –135 to +15 mV (see, e.g., DiFrancesco, 1981, p. 1986).

Data were analyzed with ClampFit10.7 (Molecular Devices).

In experiments comparing test vs. control cells to analyze the action of a drug, we incubated a fraction (normally about 50%) of the cells (test) in petri dishes with drug-containing Tyrode's solution, and the remaining fraction (control) in petri dishes with vehicle-containing Tyrode's solution. For each experiment, test and control cells were therefore always day-matched and derived from the same mouse.

## Compounds

### 5-Aminoimidazole-4-carboxamide riboside

AMPK activation was obtained by incubating SAN or HEK293 cells in Tyrode's solution containing 5-aminoimidazole-4-carboxamide riboside (AICAR), a well-known pharmacological activator of AMPK. AICAR is metabolized intracellularly to ZMP (AICAR-$PO_4$), an AMP analog (Adamovich et al., 2014). Like AMP, ZMP binds to the regulatory site of the AMPK$\gamma$ subunit, causing exposure and phosphorylation of residue Thr-172, which directly activates the kinase (Kim et al., 2016). Standard AICAR incubation concentrations in cell cultures for the molecule to exert its action as an AMPK activator are in the range 0.5–2 mM (Dagher et al., 1999; Vázquez-Chantada et al., 2010). Specifically in HEK293 cells, standard concentrations used are in the range 0.5–1 mM (Adamovich et al., 2014; Wyatt et al., 2007).

### Compound C

Compound C (dorsomorphin) is an ATP-competitive inhibitor that binds to the conserved catalytic site of the AMPK$\alpha$ subunit, thereby preventing substrate phosphorylation. Structural studies confirm this binding mode (Handa et al., 2011), and functional assays in HEK293 cells demonstrate inhibition of AMPK activity at concentrations in the range 30–40 $\mu$M (Handa et al., 2011; Thomson et al., 2008; Zhou et al., 2001).

## RT-PCR

HEK293F cells were cotransfected with pcDNA3.1_hHCN4 wild-type and GFP using TurboFect Transfection Reagent (SCR_008452; Thermo Fisher Scientific, RRID). 24 hours after transfection, cells were incubated for 4 h with AICAR (1 mM) or $H_2O$ as a control condition. Total RNA was extracted using Monarch Total RNA Miniprep Kit (New England Biolabs), quantified with a spectrophotometer (BioSpectrometer basic, Eppendorf), and diluted 1:3 in RNase-free $H_2O$. 500 ng of RNA was reverse-transcribed into cDNA using High-Capacity cDNA Reverse Transcription Kit (Applied Biosystems) in T100 Thermal Cycler (Bio-Rad) with the following temperature protocol: 25°C for 10 min, 37°C for 120 min, and 85°C for 5 min. Primers for RT-PCR were designed as follows: GAPDH Forward, 5′-CAGCCTCAAGATCATCAGCA; GAPDH Reverse, TGTGGTCATGAGTCCTTCCA; Fw pcDNA_hHCN4, 5′-CGTGGAAAGAGAGCAGGAAC-3′, Rev pcDNA_hHCN4, ATCAGGTTTCCGACCATCAG-3′. RT-PCR was performed by analyzing the fluorescence intensity of SYBR Green (Bio-Rad). Gene expression levels were evaluated by measuring the cycle threshold (Ct) and normalized to the Ct of the reference gene *GAPDH*.

## Western blot

Vehicle control HEK293F cells and cells treated with 1 mM AICAR for 4 h were lysed using heated lysis buffer (50 mM Tris-HCl, pH 6.8, 4% SDS, and 20% glycerol), and boiled at 95°C for 10 min to ensure complete protein denaturation. Lysates were centrifuged, and supernatants were stored at –20°C. Protein concentration was determined via the BCA assay using a plate reader, normalized to a BSA standard curve.

Equal protein amounts (30–40 $\mu$g) were prepared with 4× LDS sample buffer, boiled for 5 min, and separated on 12% SDS-PAGE at 120 V for 1–2 h. Proteins were transferred to a nitrocellulose membrane (0.45 $\mu$m pore size) at 70 V for 1 h on ice. Membranes were stained with Ponceau, blocked, and incubated overnight at 4°C with anti-phospho-$\alpha$1AMPK (AB_2169402; Cell Signaling, RRID) and anti-vinculin (AB_262053; Sigma-Aldrich, RRID) antibodies. After washing, HRP-conjugated secondary antibodies were applied for 1.5 h at room temperature. Signals were detected using SuperSignal West Femto Substrate and imaged with a ChemiDoc Touch system. After first analysis, the membranes were treated with western blot stripping buffer (ab282569) to remove antibodies. Then, the staining was repeated as described before to evaluate the total amount of $\alpha$1AMPK using anti-$\alpha$1AMPK (Cell Signaling). Solutions for western blot assay were the following: running buffer: 2.5 mM Tris-HCl, 19.2 mM glycine, 0.01% SDS in $H_2O$; transfer buffer: 2 mM Tris-HCl, 0.1% glycine in $H_2O$. Images were analyzed using ImageLab software (Bio-Rad), and values were normalized to the levels of the housekeeping control signal (vinculin).

## FACS analysis

24 hours after transfection, HEK293F cells were incubated 4 h with AICAR (1 mM) or $H_2O$ as control condition. Cells were detached with trypsin and resuspended in PBS + 1% BSA for blocking of nonspecific sites. Labeling with primary antibodies anti-HA-tag (Invitrogen) and isotype control IgG2bκ (Invitrogen) was performed in ice for 30 min. Goat anti-mouse IgG(H + L) Alexa 633 (Invitrogen) (AB_2535718; RRID) was used as a secondary antibody. The samples were filtered with 50/70-μm filters and analyzed with BD FACSCanto II. Lasers with a wavelength of 488 nm and an emission filter with a bandwidth of 530/30 nm, and laser with a wavelength of 633 nm and an emission filter of 660/20 nm, were used for the excitation of GFP and Alexa 633, respectively. Data were analyzed using FlowJo Single Cell Analysis Software v10 (SCR_008520; RRID). First, the GFP-positive population (FITC) expressing the construct was selected. Only in this population, the red intensity signal (APC) was measured as the geometric mean of the histogram resulting from cellular fluorescence.

## Membrane isolation

FreeStyle HEK293F cell cultures (Thermo Fisher Scientific) were transiently transfected with pEGA: HCN4ΔC (1 mg/ml) at a cell density of $2 \times 10^6$ cells per mL using polyethyleneimine (Polysciences). The transfected cells were harvested by centrifugation after 48 h of growth in shaker flasks at 37°C, 5% $CO_2$. Cell pellets were resuspended in low-salt buffer (10 mM KCl, 10 mM $MgCl_2$, 10 mM Hepes, pH 7.5, 0.5 mM PMSF, EDTA-free complete protease inhibitor cocktail [Roche] [1:1,000], 20 mg/ml DNase [Roche], and 10 mg/ml RNase [Sigma-Aldrich]) and lysed by gentle homogenization in a glass homogenizer. Membranes were isolated by ultracentrifugation (40 min at $17,000 \times g$), resuspended by homogenization, and washed twice with high-salt buffer: 1 M NaCl, 10 mM KCl, 10 mM $MgCl_2$, 10 mM Hepes, pH 7.5, 0.5 mM PMSF, EDTA-free complete protease inhibitor tablet, 20 mg/ml DNase, and 10 mg/ml RNase. Isolated membranes were resuspended by homogenization in the storage buffer: 200 mM NaCl, 20 mM Hepes, pH 7.5, 0.5 mM PMSF, and EDTA-free complete protease inhibitor cocktail (1:1,000), stored at –80°C until use.

## Protein purification in lauryl maltose neopentyl glycol/cholesteryl hemisuccinate and MS analysis

The isolated membranes were thawed on ice and solubilized by the addition of a mixture of detergents lauryl maltose neopentyl glycol (LMNG) with cholesteryl hemisuccinate (CHS) in a 5:1 ratio to a final concentration of 1% (wt/vol), and gently shaken for 2 h at 4°C. The solution was cleared by ultracentrifugation (40 min at $1,700 \times g$). Pre-equilibrated $Ni^{2+}$-NTA resin (QIAGEN) was added to the sample, together with 10 mM imidazole, and the mixture was allowed to gently rotate overnight at 4°C. After transferring the mixture to a column, the resin was washed in two steps: (1) 5 column volumes of buffer containing 50 mM imidazole; and (2) 5 column volumes of buffer containing 75 mM imidazole. The proteins were eluted with 10 column volumes of the following buffer: 200 mM NaCl, 20 mM Hepes, pH 7.5, and 300 mM imidazole. The eluted protein was loaded on a Superose 6 Increase 10/300 GL SEC column (GE Healthcare Life Sciences) pre-equilibrated with buffer containing 200 mM NaCl, 20 mM Hepes, pH 7.0, and detergent (LMNG-CHS) at the concentration of 0.002% (wt/vol). The final yield of purified protein was about 1 mg per 1 L of cells. Samples from purification were loaded in the presence and absence of DTT 50 mM on SDS-PAGE gels (NuPAGE 4–12%, Bis-Tris, Invitrogen); Novex Sharp Prestained Protein Standards (Invitrogen) were used as size references. Samples were successively stained with Coomassie blue (Sigma-Aldrich). Bands containing purified HCN4 proteins (see Fig. S4 A) were excised and analyzed by Cogentech Soc. Benefit srl with nLC-ESI-MS/MS Q Exactive HF.

## Statistical analysis

The statistical analysis of all data presented in the Results section is shown in Tables S1 and S2. The normality of data distribution in all experiments in each condition was assessed using the Shapiro–Wilk test. For V1/2 measurements, normality was a reasonable assumption and all analyses of V1/2 assumed data were normally distributed around the mean. For normalized current density data, normality was not generally a reasonable assumption owing to the skewness of the distributions and we used generalized linear models assuming a Gamma (link = "log") to fit the data (glmmTMB package, R) (see McGillycuddy et al., 2025). Following the model fit, these assumptions were assessed using residual plots.

In addition, data from isolated SAN cells used multiple cells per mouse. Potential individual mouse effects were taken into account using either a linear mixed-effects model or a generalized linear mixed-effects model (assuming a Gamma distribution). For all analyses involving pairwise comparisons when more than two groups were involved, we reported adjusted P values using Tukey's post-hoc method (emmeans package, R) (see Lenth, 2025).

All means ± SEM indicated throughout the text are the predicted means and standard errors calculated from the statistical model used to fit the corresponding data. All analyses were conducted in R version 4.4.1 (see R Core Team, 2024) or Origin (Pro) v. 2024, OriginLab Corporation (SCR_014212; RRID). R scripts and data are available upon request.

## Online supplemental material

Fig. S1 presents western blots in HEK293T cells of AMPK and pAMPK in control conditions and after AICAR treatment. In Fig. S2, the effect of AICAR on the fully activated $I_{HCN4}$ curve measured in HEK293T is compared with that measured in HEK293F cells. Fig. S3 shows FACS experiments on HEK293T cells transfected with a GFP-hHCN4-HA-tag construct. Fig. S4 shows data from an SDS-PAGE gel loaded with HCN4 proteins purified from transfected HEK293F cells under control conditions and after AICAR treatment. Table S1 shows the procedures used for the statistical analysis of $I_f$ experiments in isolated SAN cells, and Table S2 shows those used for $I_{HCN4}$ experiments in HEK293 cells.

# Results

## Pharmacological AMPK activation in primary mouse SAN cells decreases $I_f$

To verify by direct investigation of native pacemaker cells whether AMPK is a specific modulator of $I_f$, we analyzed the

effects of pharmacological activation of the kinase on $I_f$ recorded from primary SAN cells isolated from mice, by perfusing cells with the AMP activator AICAR (Kim et al., 2016).

AMPK regulation of cell metabolism can vary between males and females due to hormonal, genetic, and body composition differences, and sex-specific effects have been reported typically in connection to response to fatigue, metabolic stress, insulin resistance; furthermore, AMPK affects differently males and females in the development and progression of cardiovascular diseases (Brown et al., 2020; Kvandova et al., 2023; Strohm et al., 2025).

Although previous studies reported no significant differences in HCN4 expression between female and male mice (Yin et al., 2024), in the absence of specific information concerning sex-dependent AMPK modulation of $I_f$, we initially chose to investigate the action of pharmacological AMPK activation in SAN cells according to gender-specific protocols.

Comparison of data from different genders (Fig. 1) showed that the $I_f$ activation curve has a slightly more negative V1/2 in female-derived than male-derived cells (mean values −91.0 and −89.5 mV, respectively). Female-derived cells exhibited a lower $I_f$ amplitude compared with male-derived cells, but the difference did not reach significance (see the Fig. 1 legend).

Analysis of the effects of AMPK activation showed that AICAR treatment reduces the maximal $I_f$ density at −125 mV in both male- and female-derived cells (Fig. 1, B and D), although the reduction in female does not reach significance; neither position nor inverse-slope factor of the current activation curve was modified (Fig. 1, A and C).

These data confirm previous evidence from a study comparing mice carrying an activating mutation in the gene encoding the γ2 subunit of AMPK (*PRKAG2*) with wild-type mice (Yavari et al., 2017). At the same time, they support the view that the AMPK-dependent $I_f$ regulatory mechanism is operative in native SAN cells, and can be investigated by acute pharmacological kinase activation, without the use of transgenic models.

## Pharmacological activation of AMPK in HEK cells reduces membrane expression of HCN4 channels

The $I_f$ current is carried by HCN channels, with the HCN4 isoform being the predominant subtype in the SAN (DiFrancesco, 2020).

According to the study of Yavari et al. (2017) mentioned above, SAN cells from mice carrying a constitutively activating mutation in the gene encoding the γ2 subunit of AMPK show a markedly reduced HCN4 protein expression in the plasma membrane, and an associated decrease of $I_f$.

To verify whether pharmacological activation of AMPK does affect HCN4 membrane density in a heterologous expression setting, we investigated the behavior of HEK293T cells transfected with a plasmid carrying the human *HCN4* cDNA, both in control conditions and after 4-h incubation in the presence of the AMPK activator AICAR (1 mM).

AICAR-induced activation of the kinase in HEK293T cells was confirmed by western blot analysis of the ratio pAMPK/AMPK (Fig. S1 A).

Given that the cDNA of human HCN4 is under the control of a promoter containing TAT-box sequences, we first excluded a possible AMPK-dependent inhibition of the transcription (Sun et al., 2023). Real-time PCR showed that AICAR incubation does not alter hHCN4 mRNA expression (Fig. S1 B), implying that the effect of AMPK on the $I_f$ is posttranslational.

Patch-clamp measurements in HEK293T cells yielded results like those in SAN cells. The voltage dependence of the activation curve was not affected by AICAR treatment (Fig. 2 A), while the density of HCN4-mediated current decreased substantially (Fig. 2 B). At −130 mV, the mean $I_{HCN4}$ was −95.3 ± 12.6 ($n$ = 36) and −38.9 ± 5.29 pA/pF ($n$ = 34) in control and AICAR-treated cells, respectively. These data agree with the view that AMPK activation downregulates the membrane expression of HCN4 channels in a heterologous expression system.

Treatment with AICAR resulted in a current decrease also in HCN4-expressing HEK293F cells, a cell line used for large-scale protein purification known to have a high gene expression level (Ooi et al., 2016; Salvage et al., 2020). In a set of experiments measuring the fully activated I/V relation in control and AICAR-treated cells (Fig. S2), we found that after transfection, the mean $I_{HCN4}$ current density was higher, under control conditions, in HEK293F than in HEK293T cells (normalized conductance of 1.035 vs. 0.704 nS/pF), in agreement with a higher membrane channel expression. Also larger in HEK293F than in HEK293T cells was the fractional current reduction due to AICAR treatment (51.6 vs. 40.6%, Fig. S2). The evidence that the inhibitory action of AICAR is not reduced in highly expressing cells suggests the possibility that AMPK simply acts on any single channel protein by decreasing, with an equal probability, the chances of a successful membrane expression process.

## AMPK activation by AICAR reduces HCN4 membrane expression

To confirm that the current reduction observed in the experiments shown above in HEK cells following AICAR treatment is due to a decreased expression of HCN4 channels on the cell membrane, we performed FACS analysis of HEK293T cells transfected with GFP-hHCN4-HA-tag inserted in the S3–S4 extracellular loop, under control conditions and after AICAR treatment. The presence of the HA-tag on the channel extracellular region allowed the staining with anti-HA antibody and a secondary Alexa 633–conjugated antibody (red channel), enabling the comparison of transmembrane hHCN4 channel expression levels based on fluorescence intensity (APC). Untransfected cells were used as a negative control (negative for both FITC and APC). Transfected cells (GFP-positive, green channel) stained with isotype antibody (negative for the APC signal) were used as controls for nonspecific APC signal. As shown in Fig. S3, in AICAR-treated cells the red fluorescence intensity was on average lower compared with control cells, confirming a lower density of HCN4 channels at the plasma membrane.

## Inhibition of AMPK reverses AMPK activation–induced decrease of HCN4 current

To verify that the current reduction observed with AICAR treatment is not mediated by an unspecific, AMPK-independent

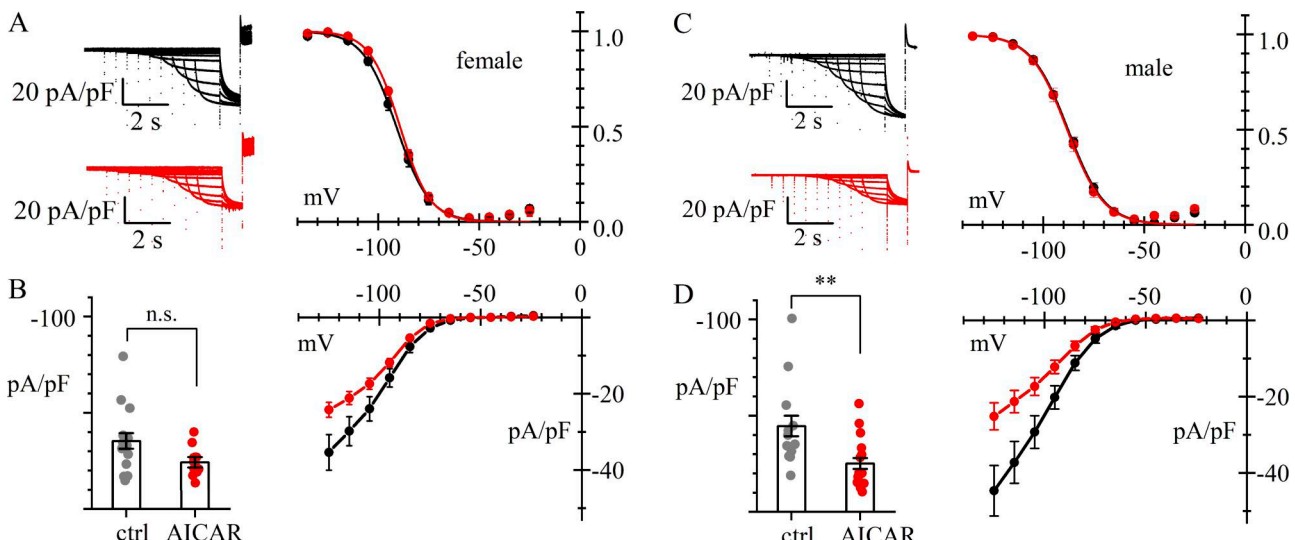

Figure 1. **AMPK activation inhibits I$_f$ in female and male mouse pacemaker cells. (A–D)** Whole-cell recordings of I$_f$ in 3-mo-old female (A and B) and 3-mo-old male (C and D) mouse SAN cells in control (black) and after 4-h treatment with AMPK activator (AICAR 1 mM) (red). Data are obtained from $N = 3$ females and $N = 3$ males. (A and C) Representative current traces (left) and activation curves (right). Best fitting with the Boltzmann equation yielded the following values: (A) V1/2 = –91.0 ± 1.67; s = 8.2 ± 0.5 mV ($n = 14$) and V1/2 = –89.5 ± 1.69 mV; s = 7.5 ± 0.4 mV ($n = 13$) (n.s.); (C) V1/2 = –88.9 ± 1.76; s = 9.2 ± 0.5 mV ($n = 12$) and V1/2 = –88.2 ± 1.65 mV; s = 9.1 ± 0.7 mV ($n = 15$) (n.s.), for control and AICAR-treated cells, respectively. (B and D) Steady-state I/V relations; bar graphs on the left show the distributions of current densities at –125 mV. Predicted means ± SEM assuming GLMM (Gamma distribution) were (B) –35.4 ± 4.03 ($n = 14$) and –24.2 ± 2.87 pA/pF ($n = 13$) (P = 0.0982, n.s.); (D) –44.6 ± 5.49 ($n = 12$) and –25.2 ± 2.77 pA/pF ($n = 15$) (P = 0.0030, **) for control and AICAR-treated cells, respectively. The difference in current density in female vs. males did not reach significance (P = 0.05067). Significance P values were adjusted with Tukey's test assuming four groups. Details of statistical analysis are shown in Table S1.

mechanism, patch-clamp experiments were conducted with the AMPK inhibitor Compound C (30 µM for 4 h) in transfected HEK293F cells. At these concentrations, Compound C efficiently inhibits AMPK activation by preventing its phosphorylation, as demonstrated by western blot analysis in previous work (Thomson et al., 2008). Untreated cells and cells incubated with AICAR (1 mM for 4 h) served as negative and positive controls, respectively.

As shown in Fig. 3 A, AICAR-treated cells (1 mM, 4 h) exhibited, as expected, a significant reduction in I$_{HCN4}$, but when treated with Compound C showed a current density like that of

untreated cells. At –135 mV, the mean current was –141.0 ± 17.6 ($n = 13$), –63.7 ± 8.7 ($n = 11$), and –158.4 ± 17.9 pA/pF ($n = 16$) in control, after AICAR treatment (P = 0.0001 vs. control, ****), and after Compound C treatment (P = 0.7688 vs. control, n.s.), respectively. This result agrees with previous data (Thomson et al., 2008) and is in accordance with the low level of phosphorylated AMPK apparent in the western blot of Fig. S1 A. Together, these data suggest that, in HEK cells, basal AMPK activity is low and insufficient to modulate HCN4 channel trafficking.

In a similar set of experiments, we applied the standard 4-h incubation protocol using either AICAR (1 mM) or the combination

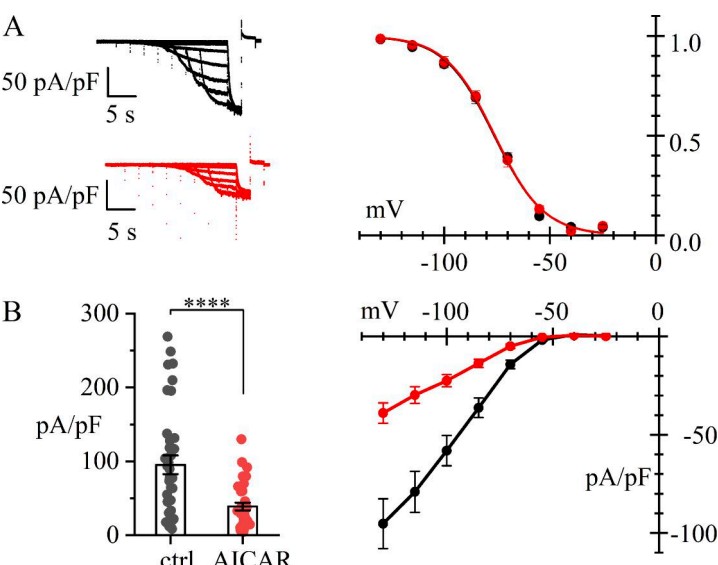

Figure 2. **AMPK activation reduces membrane expression of HCN4 channels in HEK293 cells.** Whole-cell recordings from HEK293T cells transfected with hHCN4 in control (black) and after 4-h treatment with the AMPK activator AICAR (1 mM) (red). **(A)** Representative current traces (left) and activation curves (right). Best fitting with the Boltzmann equation yielded V1/2 = –76.1 ± 1.3 ($n = 18$) and –75.0 ± 1.6 mV ($n = 16$) (unpaired $t$ test, P = 0.6105, n.s.); s = 11.8 ± 0.8 and 11.6 ± 0.6 mV, in control and AICAR-treated cells, respectively. **(B)** Steady-state I/V relations; bar graphs on the left show the distributions of current densities at –130 mV. Mean ± SEM values were –95.3 ± 12.6 ($n = 36$) and –38.9 ± 5.29 pA/pF ($n = 34$) in control and AICAR-treated cells, respectively (GLM Gamma, P < 0.0001, ****). Details of statistical analysis are shown in Table S2.

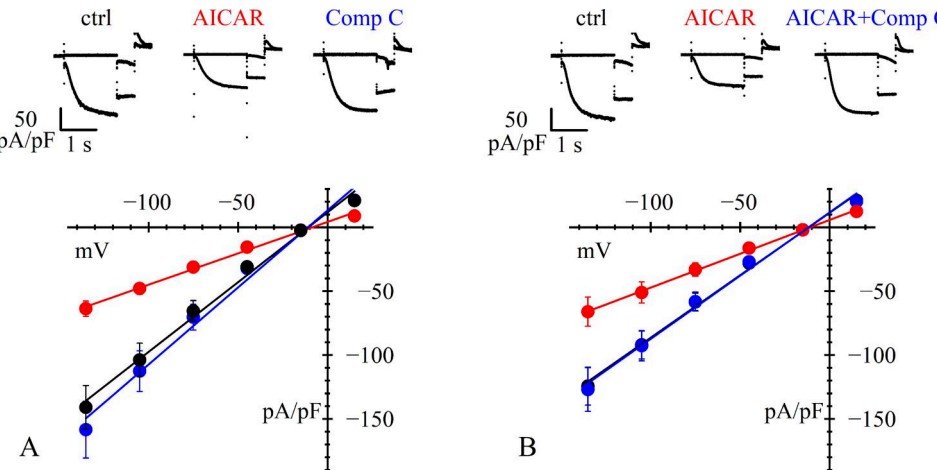

Figure 3.  **Removal of AMPK activation abolishes its action. (A and B)** Top: Representative traces of current density recorded from HEK293F cells. Data were obtained from day-matched cells in control conditions or after treatment. **(A)** Bottom: Mean fully activated I/V curves normalized to capacitance measured under control conditions (black) and after 4-h incubation with either AICAR 1 mM (red) or Compound C 30 µM (blue); normalized conductance values from linear fitting were 1.09, 0.49, and 1.21 nS/pF, respectively. **(B)** Bottom: Similar set of measurements made under control conditions (black), and after 4-h incubation with either AICAR 1 mM (red) or the combination AICAR 1 mM + Compound C 30 µM (blue); normalized conductance values were 0.98, 0.53, and 0.99 nS/pF, respectively. See Tables 1 and S2 for statistical data analysis.

AICAR + Compound C (30 µM). The results in Fig. 3 B show that cotreatment with AICAR + Compound C restores the current to control levels. At –135 mV, the current was –124.4 ± 22.2 ($n$ = 25), –66.1 ± 10.8 ($n$ = 30), and –127.0 ± 21.5 pA/pF ($n$ = 28) in control, AICAR-treated (P = 0.0243 vs. control, *), and AICAR + Compound C–treated cells (P = 0.9961 vs. control, n.s.), respectively. This confirms that the AICAR-induced current reduction is caused by AMPK activation. Statistical analysis for all data in Fig. 3 is presented in Table S2.

### Identification of a putative phosphorylation site

To investigate whether HCN4 modulation by AMPK involves direct channel phosphorylation, we searched for AMPK-consensus sites in the HCN4 sequence. Database search using Scansite 4.0 (https://scansite4.mit.edu; Obenauer et al., 2003) for AMPK-consensus motifs indeed revealed a putative consensus motive at the C terminus of hHCN4 around serine 1157 and serine 1158. This consensus is conserved in the ortholog channel RbHCN4.

Given the possibility of a direct AMPK-mediated phosphorylation of HCN4, we performed a mass spectrometry analysis of the purified proteins to evaluate which residues were phosphorylated under control conditions and after AMPK activation by AICAR treatment. With an established protocol, we therefore purified from HEK293F cells an RbHCN4 construct carrying an internal deletion that removes a poorly conserved region in the C terminus (residues 783 to 1064). The deletion was necessary to enable channel purification (Saponaro et al., 2021a; Saponaro et al., 2021b). The resulting channel retains the AMPK-consensus region and exhibits functional properties consistent with previously reported data (Saponaro et al., 2021a; Saponaro et al., 2024).

Samples of purified protein were loaded onto a diagnostic SDS-PAGE gel in the presence and absence of 50 mM DTT to favor disaggregation of the tetramers into monomers (Mishra et al., 2017). The presence of a higher amount of protein in samples treated with AICAR made it difficult to quantitatively compare samples from control and AICAR-treated cells. Nevertheless, a qualitative indication of phosphorylated residues was obtained (Fig. S4). The results reported in Fig. S4 B show that AICAR increased the overall phosphorylation of the RbHCN4 protein. Regarding specifically Ser1129 and Ser1130 (orthologs of human Ser1157 and Ser1158) in the region of the putative AMPK-consensus site mentioned above, we found that both serines were phosphorylated in AICAR samples only, supporting the hypothesis that AMPK phosphorylates HCN4 channels in this region. In the absence of AICAR, Ser1129 was found phosphorylated but not in all samples, while Ser1130 was not phosphorylated.

### The AMPK-consensus region of HCN4 is functionally involved in channel modulation by the kinase

The presence of an AMPK phosphorylation-consensus site in the HCN4 sequence raises the question whether direct channel phosphorylation is functionally relevant to the AMPK modulation of channels. To investigate this, we recorded $I_{HCN4}$ from cells transfected with phosphonull (S1157A or S1158A) and phosphomimic (S1157D or S1158D) hHCN4 mutants. The aim of this approach was to prevent serine phosphorylation (replacement with alanine) or to reproduce the phosphorylation-induced addition of a negative charge (replacement with aspartic acid), in an attempt to mimic phosphorylation itself.

Table 1 compares mean current densities (normalized to cell capacitance) recorded at –135 mV during measurement of fully activated I/V relations under control conditions and after AICAR treatment, from cells expressing wild-type or various mutant hHCN4 channels.

The data in Fig. 4 and Table 1 show that the replacement of serine 1157 by either alanine (Fig. 4 A) or aspartic acid (Fig. 4 B) makes channels insensitive to AICAR. Exposure to AICAR indeed

led, as expected, to a significant inhibition of current in wild type–transfected cells but was unable to reduce significantly the current in either S1157A- or S1157D-transfected cells. As is apparent from Fig. 4 B, the current density recorded from cells transfected with S1157D mutant channels did not appear to be any lower than, and was indeed nonsignificantly different from, that recorded from wild-type channel–transfected cells (compare current densities in Table 1). The simplest interpretation of this result is that serine replacement with aspartic acid is unable to fully mimic the effects of serine phosphorylation under basal conditions, an observation made already under various circumstances (Corbit et al., 2003; Paleologou et al., 2008).

Independently of the size of the basal current density, substitution of serine 1157 with aspartic acid made HCN4 channels unresponsive to AICAR. Together with the similar action of serine replacement with alanine, these results confirm that a serine residue at position 1157 is essential for AMPK regulation.

We next asked whether the serine residue at position 1158 could also be a target of phosphorylation by AMPK contributing to inhibition of channel trafficking/membrane expression. As shown in Fig. 5 and Table 1, we found that, as with S1157 mutants, replacement of wild-type channels with either S1158A or S1158D mutants did not significantly modify the basal current density. However, contrary to the results obtained with S1157 mutants, the expression of neither S1158A nor S1158D mutants was able to inhibit the current density reduction caused by AICAR treatment.

These results rule against the view that serine 1158 has a primary, direct role in the AICAR-induced AMPK activation and channel phosphorylation. However, in view of the evidence that this residue, too, appears to be phosphorylated by AMPK according to MS analysis, we sought to test whether it could have an ancillary action in the phosphorylating process. To do so, we analyzed the properties of the double mutant S1157D/S1158D.

As shown in Fig. 6 and Table 1, we were surprised to see that in this case, unlike the case with the single mutants S1157D or S1158D, the current density of double mutant–transfected cells under control conditions was indeed smaller than that of wild type–transfected cells, and comparable with AICAR-treated wild type–transfected cells; moreover, AICAR treatment did not further decrease the current density.

These data further reinforce the view that the AMPK-consensus region of HCN4 surrounding serine 1157 does represent a strategic site for AMPK phosphorylation and provide evidence that this is a key mechanism in the AMPK modulation of channel membrane expression. As seen in Fig. 4 above, serine 1157 must be available to AMPK-dependent phosphorylation to allow control of channel expression, but its replacement with aspartic acid is not by itself able to mimic residue phosphorylation and reduction of membrane expression. The introduction of two negatively charged residues at positions 1157 and 1158 may thus result in a stronger electrical field able to mimic phosphorylation more effectively, leading to the reduced basal current and missing effect of AMPK activation by AICAR shown in Fig. 6.

### AMPK involvement in age-related intrinsic bradycardia

We have shown that AMPK regulates $I_f$, a major factor in maintaining cardiac automaticity, specifically by phosphorylating Ser1157

Table 1. **Comparison of current densities measured in HEK293F cells**

| Fig. 4 A—GLM (Gamma) | | | |
|---|---|---|---|
| **wt** | **+AICAR** | **S1157A** | **+AICAR** |
| −133.7 ± 17.30 | −65.9 ± 6.93 | −132.9 ± 18.6 | −128.2 ± 15.80 |
| n = 20 | n = 23 | n = 17 | n = 22 |
| —***— | | —n.s.— | |
| wt vs S1157A: n.s. | | | |
| **Fig. 4 B—LM (normal)** | | | |
| **wt** | **+AICAR** | **S1157D** | **+AICAR** |
| −136.9 ± 16.3 | −68.5 ± 16.9 | −153.2 ± 15.2 | −139.2 ± 16.3 |
| n = 14 | n = 13 | n = 16 | n = 14 |
| —***— | | —n.s.— | |
| wt vs S1157D: n.s. | | | |
| **Fig. 5 A—GLM (Gamma)** | | | |
| **wt** | **+AICAR** | **S1158A** | **+AICAR** |
| −143.7 ± 13.80 | −90.4 ± 9.56 | −173.7 ± 21.20 | −106.7 ± 11.30 |
| n = 29 | n = 24 | n = 18 | n = 24 |
| —**— | | —*— | |
| wt vs S1158A: n.s. | | | |
| **Fig. 5 B—LM (normal)** | | | |
| **wt** | **+AICAR** | **S1158D** | **+AICAR** |
| −150.8 ± 9.24 | −63.3 ± 9.46 | −137.4 ± 10.60 | −75.0 ± 9.24 |
| n = 21 | n = 20 | n = 16 | n = 21 |
| —****— | | —****— | |
| wt vs S1158D: n.s. | | | |
| **Fig. 6—LM (normal)** | | | |
| **wt** | **+AICAR** | **S1157D/S1158D** | **+AICAR** |
| −156.3 ± 13.3 | −80.9 ± 15.9 | −88.7 ± 11.6 | −78.7 ± 12.8 |
| n = 13 | n = 9 | n = 17 | n = 14 |
| —**— | | —n.s.— | |
| wt vs S1157D/S1158D: ** | | | |

The table reports mean ± SEM current density measurements (pA/pF) in different conditions from experiments shown in Figs. 4, 5, and 6. Indicated are the numbers of cells (n). As indicated for each set of data, statistical analysis was performed using the generalized linear model (Gamma) or the linear model (normal) according to the criteria explained in Materials and methods. Details of the statistical analysis and the resulting significance of data comparisons are outlined for all experiments in Table S2. Data were obtained from cells expressing wild-type or mutant hHCN4 channels under control conditions or after treatment. In each experimental set, cells from the same day-matched batch were transfected with either wild-type or mutant channels, then divided into two groups, one used as control and the other exposed to AICAR.

at the C terminus of HCN4 channels. To evaluate the role of this regulatory mechanism in age-related remodeling of the heart, we compared the effects of pharmacological AMPK modulation on the $I_f$ in SAN cells from 3-mo-old (young adult) mice with that from 24-mo-old (old) mice (Figs. 7 and Fig. 8). Given the modest but significant differences observed in previous experiments (Fig. 1), only male mice, which seem to have a larger $I_f$ and a larger reduction after AMPK activation, were analyzed.

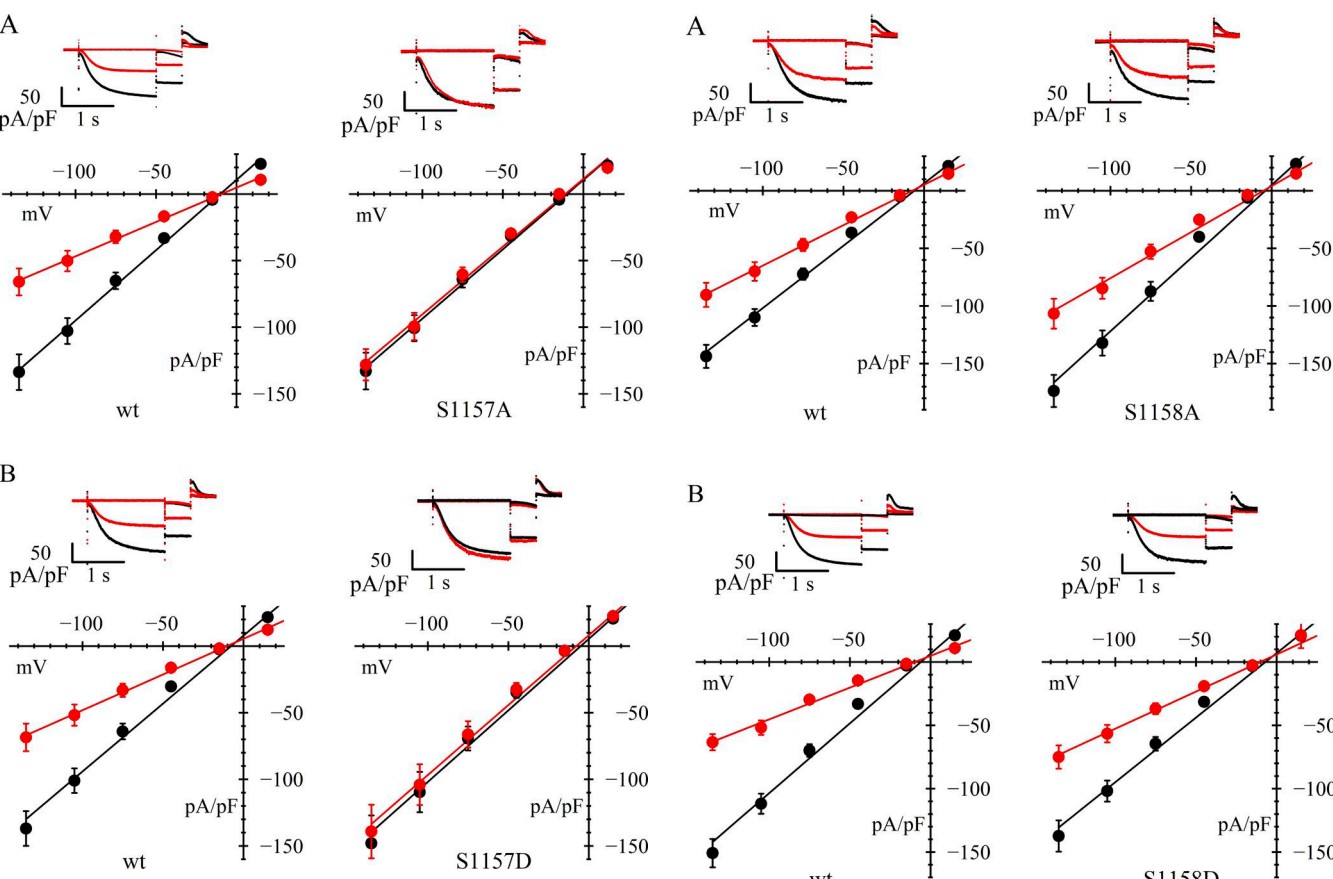

Figure 4. **Involvement of hHCN4 serine 1157 in AMPK-mediated channel modulation. (A and B)** In A and B, upper panels are typical current records, and lower panels are fully activated I/V relations, normalized to cell capacitance, from HEK293F cells in control conditions (black) and after AICAR treatment (red). Data from cells transfected with wild-type hHCN4 (left in A and B) are compared with data from cells transfected with S1157A or S1157D mutant channels (right in A and B, respectively). Linear fitting of I/V curves yielded the following normalized conductance values (nS/pF): (A) 1.06, 0.52 (left) and 1.04, 1.01 (right); (B) 1.02, 0.54 (left) and 1.08, 1.05 (right) for wild-type and AICAR-treated cells, respectively. In each experiment, control and mutant channels data are from the same day-matched transfection protocol. Statistical analysis and data comparisons are shown in Tables 1 and S2.

Figure 5. **hHCN4 serine 1158 alone does not directly contribute to AMPK-mediated channel modulation. (A and B)** In A and B, upper panels are typical current records, and lower panels are fully activated I/V relations, normalized to cell capacitance, from HEK293F cells in control conditions (black) and after AICAR treatment (red). Data from cells transfected with wild-type hHCN4 (left in A and B) are compared with data from cells transfected with S1158A or S1158D mutant channels (right in A and B, respectively). Linear fitting of I/V curves yielded the following normalized conductance values (nS/pF): (A) 1.08, 0.70 (left) and 1.27, 0.80 (right); (B) 1.10, 0.50 (left) and 1.02, 0.59 (right) for wild-type and AICAR-treated cells, respectively. Statistical analysis and data comparisons are shown in Tables 1 and S2.

Consistent with the intrinsic bradycardia typical of elderly subjects, 24-mo-old mice exhibited reduced $I_f$ density at −125 mV compared with 3-mo-old mice. No differences were observed in the $I_f$ voltage activation curves between the two groups, with mean half-activation voltages of −89.5 in old mice and −89.9 mV in young mice. As described in Fig. 7, B and D, while AMPK activation significantly reduced $I_f$ density at −125 mV in 3-mo-old mice, no such effect was observed in 24-mo-old mice. In those mice, untreated cells displayed the same current amplitude as AICAR-treated cells, suggesting a constitutive activation of the kinase in this group.

**AMPK inhibition rescues $I_f$ in old mice**
The evidence in Fig. 7 that in old mice $I_f$ is insensitive to AMPK activation, along with the evidence that the basal density in control conditions is lower than that recorded in the young mice, suggests constitutive kinase activation. However, an

age-related decline in kinase responsiveness cannot be excluded. To investigate this possibility, patch-clamp experiments were performed in young and old mice to compare $I_f$ before and after treatment with the AMPK inhibitor Compound C (30 μM for 4 h).

As shown in Fig. 8, A and B, cells from 3-mo-old mice showed similar $I_f$ density at −125 mV under control conditions and following treatment with Compound C (Fig. 8 B), with no modification of activation curve parameters (Fig. 8 A), in agreement with the view that in the young mice, basal AMPK activity is too low to affect the membrane density of f-channel expression. On the contrary, as shown in Fig. 7, C and D, in 24-mo-old mice, though still in the absence of any modification of the $I_f$ activation curve (Fig. 8 C), the $I_f$ density at −125 mV increased significantly after AMPK inhibition (Fig. 8 D), strongly supporting constitutive AMPK activation in cells from aged animals.

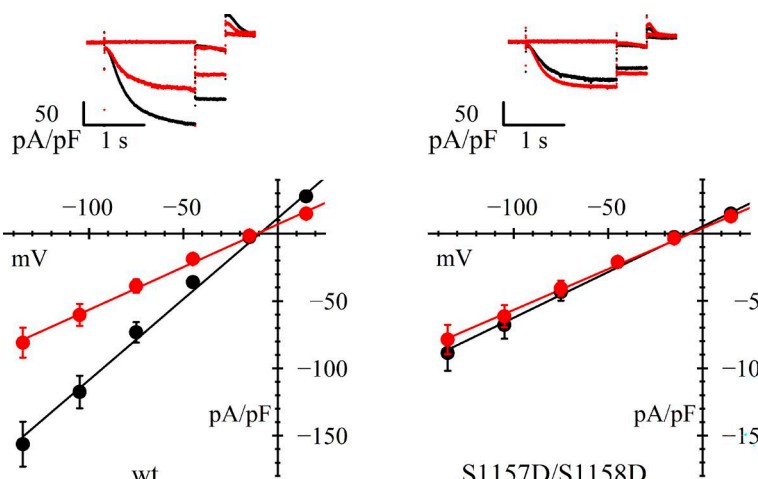

Figure 6. **Loss of AMPK action on double mutant S1157D/S1158D.** Representative current records (top) and fully activated I/V relations (bottom) normalized to cell capacitance, from HEK293F cells in control conditions (black) and after AICAR treatment (red). While cells transfected with wild-type hHCN4 (left) show a standard response to AICAR treatment, cells transfected with S1157D/S1158D double-mutant channels in day-matched experiments (right) have a reduced basal current and are unresponsive to AICAR. Linear fitting of I/V curves yielded the following normalized conductance values (nS/pF): 1.20, 0.63 (left) and 0.68, 0.60 (right) for wild-type and AICAR-treated cells, respectively. Statistical analysis and data comparisons are shown in Tables 1 and S2.

## Discussion

AMPK is a heterotrimeric serine/threonine kinase comprising a catalytic α and two regulatory β and γ subunits, the latter existing as three isoforms in mammals (Oakhill et al., 2011). Previous work has shown that constitutive activation of AMPK caused by mutations in *PRKAG2*, the gene coding for the γ2 subunit, leads to a spectrum of cardiac disorders, among which a prominent one is bradycardia (Arad et al., 2003; Blair et al., 2001; Davies et al., 2006; Gollob et al., 2001; Yavari et al., 2017). Coherent with the function of $I_f$ in pacemaker rate regulation, studies of genetic mouse models exploring the cellular basis of AMPK-dependent modulation of SAN activity have indeed shown that among the cellular factors affected by AMPK activation, a major role in

bradycardia is played by a decrease in membrane density of funny channels (Yavari et al., 2017).

These data support a role of AMPK, for example, in the intrinsic resting bradycardia that develops in athletes following endurance exercise activity (D'Souza et al., 2014; Yavari et al., 2017). But an even more basic function proposed for AMPK, in the framework of long-term cardiac homeostasis, is the metabolic regulation of HR at a cellular level, operated according to a negative feedback mechanism able to balance cardiac work with energy supply.

This negative feedback safety mechanism works based on the known direct proportionality between cardiac oxygen consumption and HR (Boerth et al., 1969). Thus, through AMP-dependent AMPK activation, when energy availability decreases, slowing of

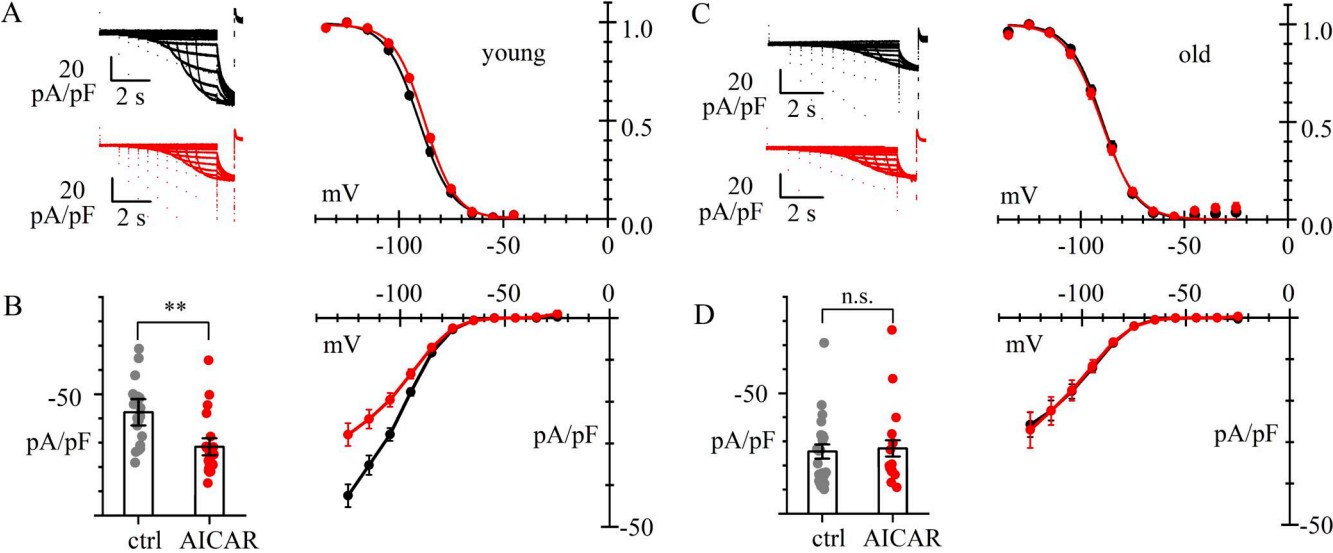

Figure 7. **Age dependence of AMPK action of $I_f$ in pacemaker cells. (A–D)** Whole-cell recordings of $I_f$ in 3-mo-old (young) (A and B) and 24-mo-old (old) (C and D) mouse SAN cells in control (black) and after 4-h treatment with AMPK activator (AICAR 1 mM) (red). Data are obtained from $N = 3$ young and $N = 4$ old mice. **(A and C)** Representative current traces (left) and activation curves (right). Best fitting yielded the following values (mV): (A) V1/2 = −89.9 ± 0.81 ($n = 21$) and −87.8 ± 0.88 ($n = 18$) (n.s.); s = 8.0 ± 0.4 and 7.4 ± 0.3, for controls and AICAR-treated cells, respectively; (C) V1/2 = −89.5 ± 0.78 ($n = 23$) and −90.3 ± 0.97 ($n = 15$) (n.s.); s = 8.0 ± 0.4 and 8.5 ± 0.7 for control and AICAR-treated cells, respectively. **(B and D)** I/V relations; bar graphs on the left show the distributions of current densities at −125 mV. Mean ± SEM values were (pA/pF) as follows: (B) −42.8 ± 5.34 ($n = 21$) and −27.9 ± 3.48 ($n = 21$) (P = 0.0037, **); (D) −25.4 ± 2.91 ($n = 23$) and −26.5 ± 3.37 ($n = 16$) (P = 0.9894, n.s.), for control and AICAR-treated cells, respectively. The difference between current densities from old and young mice in control conditions was significant (P = 0.0112, *). Details of statistical analysis are shown in Table S1.

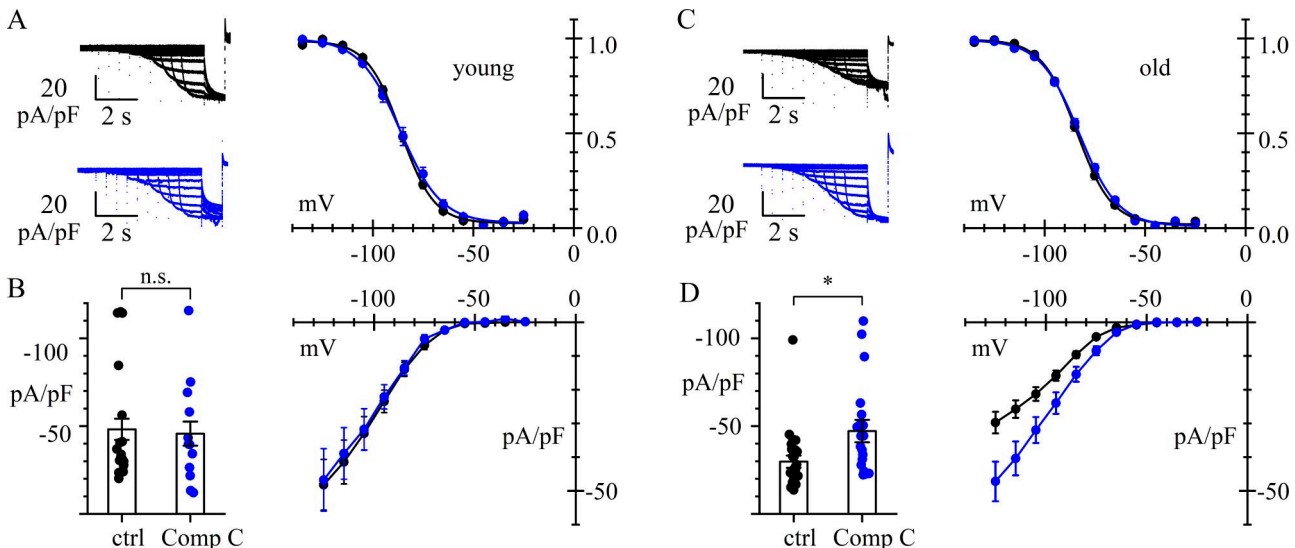

**Figure 8.** **AMPK is constitutively activated in old, but not in young, mice. (A–D)** Whole-cell recordings of $I_f$ in 3-mo-old (young) (A and B) and 24-mo-old (old) (C and D) mouse SAN cells in control (black) and after 4-h treatment with AMPK inhibitor (Compound C 30 μM for 4 h) (blue). Data are obtained from $N = 3$ young and $N = 5$ old mice. **(A and C)** Representative current traces (left) and activation curves (right). Best fitting yielded the following values (mV): (A) V1/2 = $-86.2 \pm 1.30$ ($n = 18$) and $-86.4 \pm 1.42$ ($n = 14$) (n.s.); s = 8.3 ± 0.4 and 10.2 ± 0.7; (C) V1/2 = $-83.6 \pm 1.04$ ($n = 24$) and $-82.5 \pm 1.19$ ($n = 17$) (n.s.); s = 8.7 ± 0.3 and 9.6 ± 0.4, for control and Compound C–treated cells, respectively. **(B and D)** I/V relations; bar graphs on the left show the distributions of current densities at $-125$ mV. Mean ± SE values (pA/pF) were as follows: (B, young) $-45.3 \pm 6.94$ ($n = 19$) and $-44.1 \pm 7.24$ ($n = 16$) ($P = 0.9986$, n.s.); and (D, old) $-29.1 \pm 3.55$ ($n = 27$) and $-47.1 \pm 6.37$ ($n = 20$) ($P = 0.0110$, *), for control and Compound C–treated cells, respectively. Current densities from young mice in control conditions and from old mice after Compound C treatment did not differ significantly ($P = 0.9975$, n.s.). Details of statistical analysis are shown in Table S1.

HR caused by a generalized remodeling, including reduced $I_f$ channel membrane expression, acts to limit expenditure, in an attempt to reestablish a correct equilibrium between energy supply and energy consumption.

Here, we have investigated the mechanism underlying AMPK-dependent reduction of $I_f$ and related pacemaker rate reduction.

Our findings reveal that activation of AMPK by AICAR leads to phosphorylation of HCN4 at Ser1157 within its C-terminal region, resulting in a reduction of the channel's membrane expression. Notably, this residue corresponds to Ser1154 described by Liao et al. (2010), who identified a cluster of threonine and serine residues (Thr1153, Ser1154, Ser1155, and Ser1157 in their numbering) as critical for PKA-dependent modulation of HCN4. This region therefore appears to represent a key hotspot for kinase-mediated regulation.

In their study, inhibition of PKA with a specific inhibitory peptide abolished the β-adrenergic–induced depolarizing shift in the voltage dependence of activation, indicating that phosphorylation within this cluster mediates rapid cAMP-dependent modulation of the channel.

The mechanism we describe differs from that of Liao et al. (2010). Our results identify Ser1157 (corresponding to Ser1154 in Liao et al. [2010]) as a specific target for AMPK-dependent phosphorylation, which acts over longer timescales by reducing HCN4 surface expression rather than altering gating properties. Considering that AMPK is activated during energy deprivation, phosphorylation of this site may serve as a molecular switch integrating metabolic status with pacemaker activity, thereby maintaining energetic homeostasis in cardiac automaticity. Furthermore, this mechanism explains the observed decrease in $I_f$

following the pharmacological activation of AMPK in isolated mouse SAN cells (Figs. 1, 7, and 8).

Our data provide a mechanistic explanation of the process underlying the pathological bradycardia reported in patients carrying a gain-of-function mutation in the γ-2 subunit of AMPK (Yavari et al., 2017). Moreover, they provide evidence that this regulatory pathway contributes to the intrinsic age-related sinus bradycardia (Jose and Collison, 1970; Larson et al., 2013; Monfredi and Boyett, 2015; Peters et al., 2020). In support of this, we show that SAN cells from elderly animals exhibit a reduced $I_f$ density compared with that recorded from young animals. We also show that in aged mouse pacemaker cells, pharmacological AMPK activation does not further decrease $I_f$ density, while on the contrary, AMPK inhibition restores $I_f$ levels to values comparable to those in young animals.

Our results agree with previous studies in rodents showing that a downregulation of HCN transcripts is associated with age-dependent slowing of intrinsic HR (Huang et al. 2007; Huang et al., 2016; Tellez et al., 2011). In other studies, a role of $I_f$ in the age-dependent reduction in pacemaker function was proposed based on evidence that in aged mice, the $I_f$ activation curve shifts to more negative voltages, leading to a reduced contribution of the $I_f$ to pacemaker activity (Larson et al., 2013; Sharpe et al., 2017). Although we did not specifically investigate age-related changes of the $I_f$ properties, our data do not seem to indicate that the voltage dependence of activation is significantly modified by age (Figs. 7 and 8). We have no immediate explanation for these differences, except to notice differences in the voltage-clamp protocol and intracellular and extracellular solutions.

It may be interesting to note that Larson et al. (2013) also recorded a diminished current density in older mice. The percent

reduction in $I_f$ density when comparing 2–3 mo with 21–24-mo-old mice (34% reduction from –27.7 to –18.2 pA/pF, (Larson et al., 2013; and Table 3 in Larson et al., 2013) was close to the values found in our results (39% in Fig. 7, 38% in Fig. 8).

AMPK has a key role in cellular energy homeostasis, and its activation will likely affect also pathways other than the membrane expression of HCN4 channels. However, we have no evidence for the involvement of other AMPK-phosphorylated proteins in the AMPK modulation of HCN4; in fact, the demonstration of two opposite effects on HCN4 of the activation/inhibition of the kinase by AICAR and Compound C, respectively, is strong evidence supporting a specific AMPK-dependent control of HCN4 expression.

This work proposes that AMPK controls trafficking and/or expression of funny channels on the membrane of SAN pacemaker cells by phosphorylation of a specific consensus site in the C terminus of HCN4 proteins but does not resolve the molecular basis of the mechanism by which AMPK-dependent channel phosphorylation modulates expression. Channel expression on the membrane is the final stage in a complex set of processes starting from protein synthesis and involving posttranslational modifications in the ER, packaging, Golgi processing, trafficking, and final insertion in the membrane, and many of these steps may be affected by AMPK-dependent phosphorylation. A limitation of our data is that they do not allow to distinguish which among the several molecular/cellular steps potentially involved is the one mediating the AMPK-dependent action. Further investigation will be required to elucidate the specific cellular location and mechanism responsible for the reduced membrane expression of HCN4 channels.

Overall, our data support the hypothesis that under normal physiological conditions, AMPK activity is age-related: it is low or nil in young animals, but becomes progressively higher in old animals, according to a mechanism that contributes to the age-dependent decline in intrinsic pacemaker rate (Larson et al., 2013; Ostchega et al., 2011) via downregulation of membrane expression of HCN4 channels in cardiac pacemaker cells (Monfredi and Boyett, 2015).

Aging is the most common cause of sinus node disease. Thus, a deeper understanding of the pathway underlying long-term regulation of intrinsic HR by AMPK-dependent HCN4 channel phosphorylation and associated slowing of HR in the elderly can help provide insight into preventing electronic pacemaker implantation and mitigating cardiac complications associated with age.

### Data availability

The data are available from the corresponding authors upon reasonable request (dario.difrancesco@unimi.it; luca.palloni@unimi.it).

## Acknowledgments

Jeanne M. Nerbonne served as editor.

This work was supported by the Fondation Leducq Research Grant TNF FANTASY 19CVD03 to D. DiFrancesco and A. Moroni (Milano) and M.E. Mangoni (Montpellier).

Author contributions: Luca M.G. Palloni: conceptualization, data curation, formal analysis, investigation, methodology, and writing—original draft, review, and editing. Nicole Sarno: conceptualization, formal analysis, investigation, and methodology. Caterina Azzoni: formal analysis, investigation, and methodology. Nicol Furia: formal analysis, investigation, and methodology. Matteo E. Mangoni: conceptualization, funding acquisition, project administration, and supervision. Alessandro Porro: conceptualization, data curation, investigation, methodology, and writing—review and editing. Teresa Neeman: formal analysis and writing—review and editing. Andrea Saponaro: conceptualization and writing—review and editing. Gerhard Thiel: supervision and writing—review and editing. Anna Moroni: conceptualization, supervision, and writing—review and editing. Dario DiFrancesco: conceptualization, formal analysis, funding acquisition, investigation, methodology, project administration, resources, supervision, and writing—original draft, review, and editing.

Disclosures: The authors declare no competing interests exist.

Submitted: 17 August 2025

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

# Supplemental material

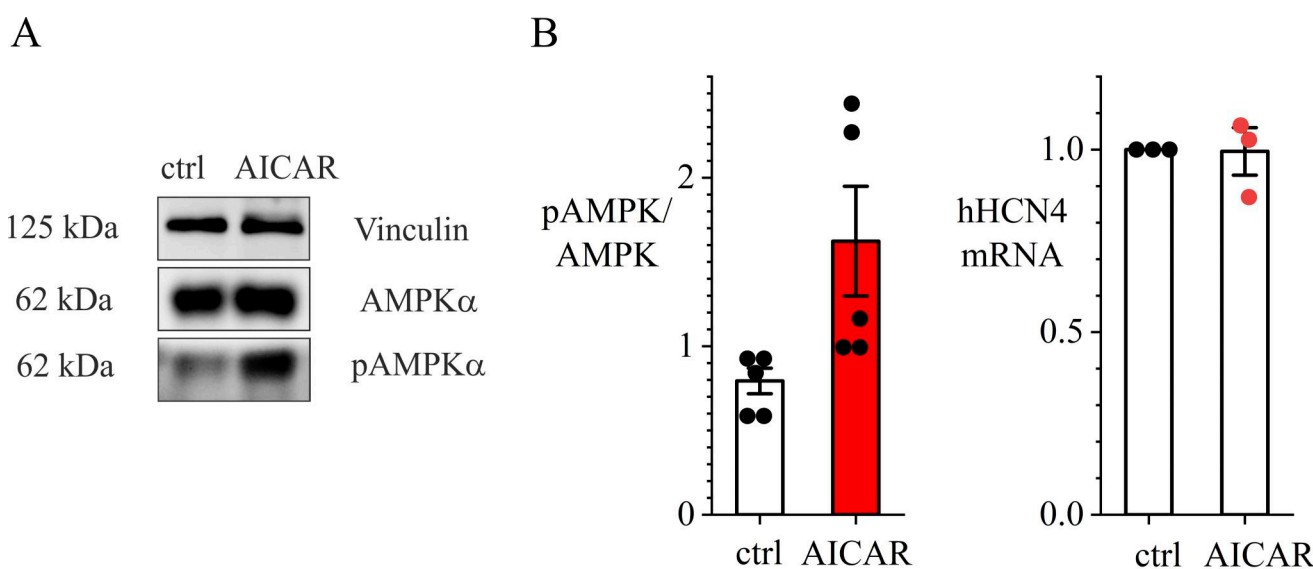

Figure S1. **Western blot analysis of the ratio pAMPK/AMPK. (A)** Left: Representative western blot of AMPK and pAMPK in HEK293T cells under control conditions and following 4-h incubation in AICAR-containing solution (1 mM). Right: Quantification of active AMPK (pAMPK) protein expression related to the total AMPK (unpaired *t* test, P = 0.038, *n* = 5). **(B)** Real-time PCR from HEK293T cells transfected with hHCN4 from three technical replicates, indicating no significant fractional reduction in hHCN4 mRNA levels after AICAR treatment (paired *t* test, P = 0.5238, n.s.). Source data are available for this figure: SourceData FS1.

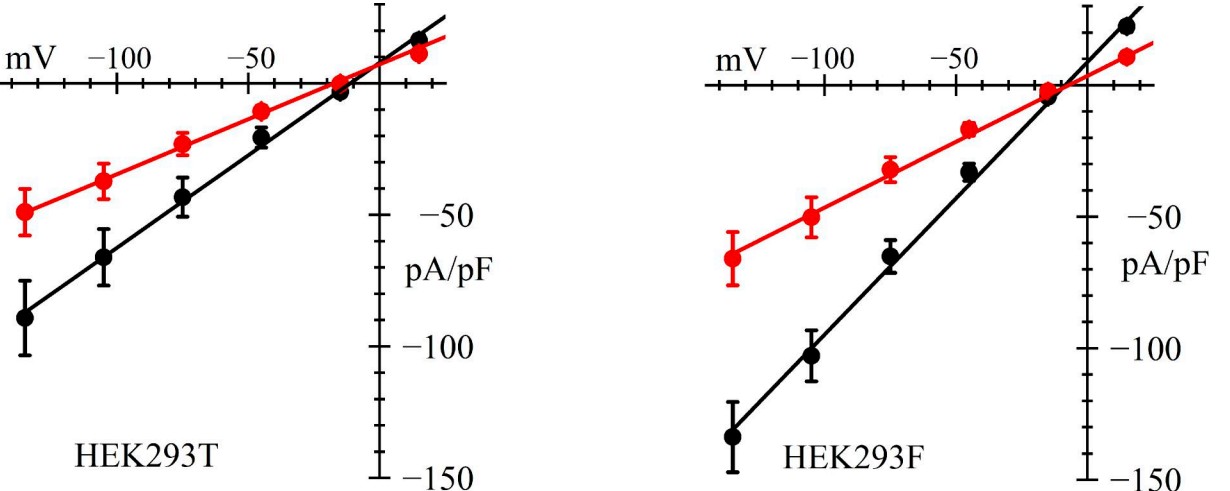

**Figure S2.  AICAR-induced I$_{HCN4}$ inhibition in HEK293T vs HEK293F cells.** Fully activated I$_{HCN4}$ I/V curves normalized to capacitance measured from hHCN4-transfected HEK293T (left) or HEK293F cells (right) under control conditions (black) and following 4-h incubation in AICAR-containing solution (1 mM) (red). Linear fitting led to normalized conductance values of 0.704 ($n$ = 16) and 0.418 nS/pF ($n$ = 13) for HEK293T and 1.035 ($n$ = 20) and 0.501 nS/pF ($n$ = 23) for HEK293F cells, in control conditions and following AICAR incubation, respectively. The current density reduction due to AICAR treatment was thus 40.6% in HEK293T and 51.6% in HEK293F cells.

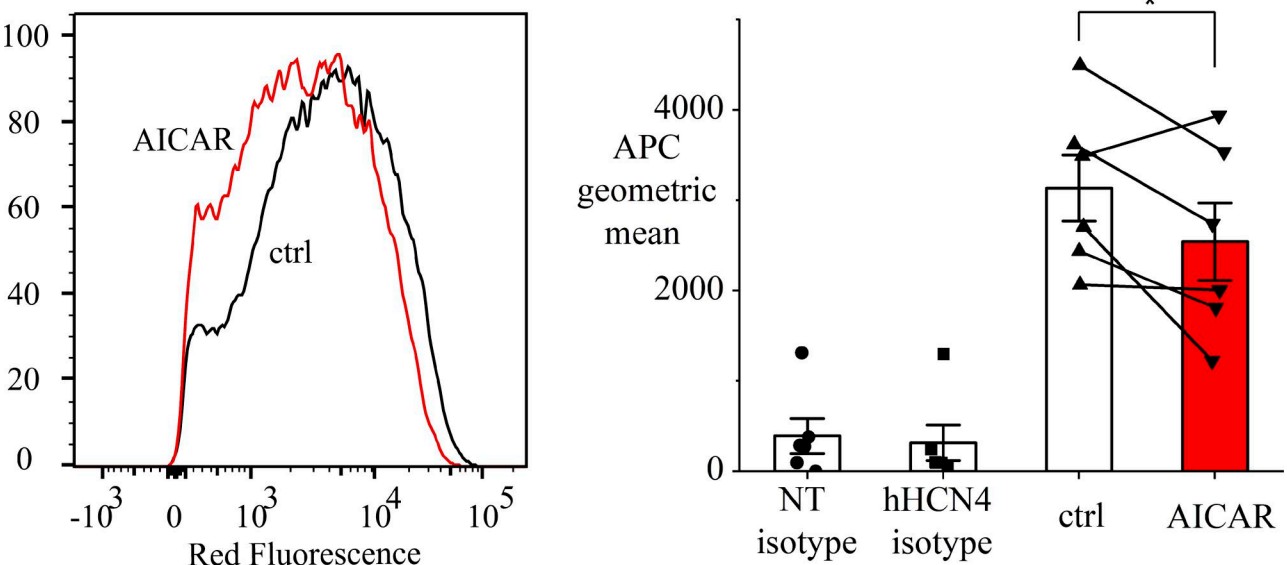

**Figure S3.  FACS experiments on HEK293T cells transfected with a plasmid carrying the construct GFP-hHCN4-HA-tag.** The green GFP signal reports in this case the synthesis of hHCN4 and the red HA-tag the positive insertion of the channel in the plasma membrane. Left: Representative histogram of normalized red fluorescence intensity signal in green positive population. Right: Plot of the geometric means of six technical replicates. APC geometric means were 3,135 ± 367 in control runs, and 2,542 ± 430 after AICAR treatment (paired $t$ test, P = 0.044).

# A

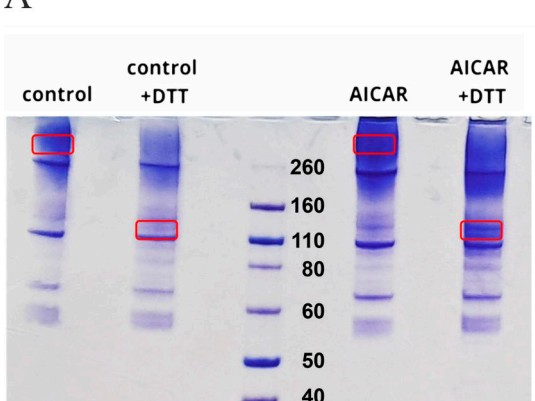

| | | | |
|---|---|---|---|
| control | control +DTT | AICAR | AICAR +DTT |

260
160
110
80
60
50
40

# B

| CTRL | CTRL + DTT | AICAR | AICAR + DTT |
|---|---|---|---|
| CTRL | CTRL +DTT | AICAR | AICAR + DTT |
| | | S59 | S59 |
| | | S120 | S120 |
| | | T121 | T121 |
| | | S125 | S125 |
| | | S131 | S131 |
| S720 | S720 | S720 | S720 |
| S1090 | S1090 | S1089 | S1089 |
| | | S1092 | S1092 |
| | | S1093 | S1093 |
| | | S1094 | S1094 |
| | | S1097 | S1097 |
| S1129 | | S1129 | S1129 |
| | | S1130 | S1130 |
| | | S1132 | |

Figure S4. **SDS-Page analysis of AICAR action on HCN4 protein phosphorylation. (A)** SDS-PAGE gel loaded with HCN4 proteins purified from transfected HEK293F cells under control conditions and after AICAR treatment. The quaternary structure of HCN4 is not completely dissolved in a denaturing gel (Saponaro et al., 2021a; Saponaro et al., 2021b). Therefore, DTT was added to favor the disruption of HCN4 tetramers and the consequent isolation of the monomers. Circled bands were analyzed with nLC-ESI-MS/MS Q Exactive HF. **(B)** Table of all residues found to be phosphorylated in MS analysis. Source data are available for this figure: SourceData FS4.

**Provided online are Table S1 and Table S2. Table S1 shows data from $I_f$ experiments in isolated SAN cells. Table S2 shows data from $I_{HCN4}$ experiments in HEK293 cells.**

