## [Peer Review File · The Journal of General Physiology]

AMPK-mediated HCN4 channel phosphorylation contributes to age-related intrinsic bradycardia

Luca Palloni, Nicole Sarno, Caterina Azzoni, Nicol Furia, Matteo Mangoni, Alessandro Porro, Teresa Neeman, Andrea Saponaro, Gerhard Thiel, Anna Moroni, and Dario DiFrancesco

Corresponding Author(s): Dario DiFrancesco, University of Milan and Luca Palloni, University of Milan

Review Timeline:

Submission Date:	August 17, 2025
Editorial Decision:	September 24, 2025
Revision Received:	December 8, 2025
Editorial Decision:	December 21, 2025
Revision Received:	December 29, 2025

Editor: Jeanne Nerbonne

Transaction Report:

DOI: <https://doi.org/10.1085/jgp.202513873>

September 24, 2025

Professor Dario DiFrancesco
University of Milan
Department of Biosciences
via Celoria 26
Milano I-20133
Italy

Re: 202513873

Dear Dario,

Thank you for submitting your manuscript, titled "AMPK-mediated HCN4 channel phosphorylation contributes to age-related bradycardia," to the Journal of General Physiology (JGP). Your manuscript has now been seen by three (3) experts in the field; their comments are appended below. As you will see, the reviewers are quite enthusiastic about the work. They have, however, also raised a number of conceptual and experimental concerns that the Editors agree should be addressed prior to further consideration of the manuscript for publication in JGP. With regard to the reviewers' request for clarification of the number of cells (n) and number of animals (N) studied in the experiments presented in Figures 1, 7, and 8, the Editors noted that the (control) data do not appear to be normally distributed, raising concern about the use of an unpaired T test to compare the control and drug-treated cells. Also, there is no mention of (or way to evaluate from the presentation) whether the variances were the same in the two groups and/or among the cells studied in animals. Alternative statistical methods (such as nested T tests) should be used to evaluate the cell/animal data presented. In re-analyzing these (animal experiment data), it might also be useful to evaluate the normalcy of the HEK cell data sets presented in Figure 2, as well as in all of the data sets used to generate the other figures (3,4,5, and 6) in which cumulative electrophysiological data are presented.

We would be pleased to receive a suitably revised manuscript that addresses the identified concerns. Please be aware that we will have your revised manuscript re-reviewed, preferably by the original reviewers, pending their availability. Also, please do not hesitate to contact me (via the editorial office) if you feel that a discussion of this letter and/or the reviewers' comments would be helpful.

Please submit your revised manuscript via the link below along with a point-by-point letter that details your responses to this letter and to the detailed comments of the 3 reviewers, as well as a copy of the text with alterations highlighted (boldfaced or underlined). If the article is eventually accepted, it would include a 'revised date' as well as submitted and accepted dates. If we do not receive the revised manuscript within one year, we will regard the article as having been withdrawn. We would be willing to receive a revision of the manuscript at a later time, but the manuscript will then be treated as a new submission, with a new manuscript number.

Please pay particular attention to recent changes to our instructions to authors in the following sections: Data presentation, Blinding and randomization and Statistical analysis, under Materials and Methods, as shown here: <https://rupress.org/jgp/pages/submission-guidelines#prepare>. Re-review will be contingent on inclusion of the required information (including for data added during revision) and demonstration of the experimental reproducibility of the results. Also, to improve the reproducibility of published content, we have partnered with SciScore. Authors are prompted in eJP to copy and paste the Materials and Methods section of their manuscript for a SciScore assessment when submitting their revised manuscript. Authors are encouraged (not required) to further revise their Materials and Methods if the SciScore is below 4. More information can be found here: <https://rupress.org/jgp/pages/submission-guidelines#sciscore>

Please note, JGP requires authors to submit Source Data used to generate figures containing gels and Western blots with all revised manuscripts (when applicable). This Source Data consists of fully uncropped and unprocessed images for each gel/blot displayed in the main and supplemental figures. If your paper includes cropped gel and/or blot images, please be sure to provide one Source Data file for each figure that contains gels and/or blots along with your revised manuscript files. File names for Source Data figures should be alphanumeric without any spaces or special characters (i.e., SourceDataF#, where F# refers to the associated main figure number or SourceDataFS# for those associated with Supplementary figures). The lanes of the gels/blots should be labeled as they are in the associated figure, the place where cropping was applied should be marked (with a box), and molecular weight/size standards should be labeled wherever possible. Source Data files will be made available to reviewers during the evaluation of revised manuscripts, and if your paper is eventually published in JGP, the files will be directly linked to specific figures in the published article.

Source Data Figures should be provided as individual PDF files (one file per figure). Authors should endeavor to retain a minimum resolution of 300 dpi or pixels per inch. Please review our instructions for export from Photoshop, Illustrator, and PowerPoint here: <https://rupress.org/jgp/pages/submission-guidelines#revised>

When revising your manuscript, please be sure it is a double-spaced MS Word file and that it includes editable tables, if appropriate.

Please submit your revised manuscript via this link:
Link Not Available

Thank you for submitting this manuscript to JGP.

Sincerely,

Jeanne

Jeanne Nerbonne, Ph.D.
On behalf of the Journal of General Physiology

Journal of General Physiology's mission is to publish mechanistic and quantitative molecular and cellular physiology of the highest quality; to provide a best-in-class author experience; and to nurture future generations of independent researchers.

Reviewer #1 (Comments to the Authors):

In this paper, the action of AMPK on the funny channel of the sinoatrial node is examined. It was shown previously that AMPK activation leads to bradycardia and a reduction of HCN4 at the cell surface of pacemaker cells. Here, it is further shown that the reduction may be linked to direct phosphorylation of specific C-terminal residues of HCN4. Another novel finding is that AMPK may be constitutively active and maintains funny channels at a reduced level in older mice. This could explain the slowing of heart rate in older people in which AMPK may be more active because of changes in metabolism including reduced availability of ATP.

The functional data are very clear, easy to follow and comprehensive, and support the idea that AMPK regulates the surface expression of funny channels. The biochemical data is also very clear and provide support for most of the observations (but not all, see below). I think that a strength of the approach is the ability quantitatively analyze the electrical and biochemical data related to HCN4 and AMPK. However, the cellular and molecular process by which the funny current and channel protein at the plasma membrane is increased is not clear. This is also mentioned in the Discussion and represents limitation of the approach.

Pharmacological activator and inhibitor of AMPK are used which presents both a weakness and a strength. The weakness is that these are used at very high concentrations and therefore it seems at least possible that they are having effects on proteins other than the HCN4 protein and these could be involved in the effects observed. Also, even very specific actions on AMPK may lead to changes in phosphorylation of proteins other than the funny channel. The strength of the use of both drugs is that they produce roughly opposite effects on funny current as would be expected by activation and inhibition of AMPK in both sinoatrial node cells and HEK cells.

Another strength of this study is that the effects of these drugs do not alter the current activation curve. The whole-cell measurements are carried out without any cAMP in the pipette. Thus, the increase and decrease in the current is due to an effect specifically on the fully-activated current. In particular, this rules out an effect of current rundown due to a negative shift of the activation curve in the AICAR-induced decrease in current.

A possible weakness is that the effects of the drugs in HCN4-expressing HEK cells may not reflect the process in sinoatrial node cells. However, in HEK cells, a decrease in funny current upon exposure to AICAR (AMPK activator) and the absence of any effect on the current activation curve suggests that AMPK is acting in a similar manner in both cell types. Likewise, lack of effect when AICAR and Compound C are combined in HEK cells follows a similar logic to what is thought to occur in sinoatrial cells. Thus, it seems reasonable to suggest that the regulation of funny channels by AMPK in the sinoatrial pacemaker cells and HEK cells is similar.

I have some specific comments below.

Fig.1. AMPK activation inhibits I_f in female and male mice pacemaker cells.

These experiments show that funny current is reduced by AMPK activation. Activation of AMPK is carried out by exposing cells to AICAR or control for four hours. The activation curves are not significantly affected by this drug. The current, at fully-activated voltages, is significantly larger after AMPK activation.

The reduction of the funny current by AICAR is clear. However, the concentration of AICAR, at 1 mM, seems very high. What is the nature of this drug, how does it work and why is it used at such high concentrations?

In later figures, AICAR is shown to not decrease the funny current in older mice. How old were the mice in Figure 1?

The use of n=4 female and n=3 male mice is helpful. But it might be easier to follow if a different letter is used for number of animals (n) versus number of cells (also n).

I assume that different sets of cells are used to compare the drug with control given the long exposure time (true?).

Fig. 7. Age-dependence of AMPK action of If in pacemaker cells SAN Cells.

These experiments show that the funny current is decreased by AMPK activation (in young (3 months) but not old (6 months) mice.

Are the mice male or female?

Fig. 8. AMPK is constitutively activated in old but not in young mice.

These experiments show that the funny current is increased in old (6 months) but not young (3 months) mice by AMPK inhibition (Compound C 30 μ M for 4 hours). Together, these data suggest that AMPK is constitutively activated in older mice.

The increase in funny current of young mice, by Compound C, is also clear. However, even though the concentration is lower than the concentration of AICAR, at 30 M it also seems high. What is the nature of Compound C, how does it work and how was this concentration decided upon?

Are the mice male or female?

Fig. 2. AMPK activation (AICAR) reduces membrane expression of HCN4 channels in HEK293 cells. Fig. 3. Removal of AMPK activation abolishes its action.

AICAR (stimulation of AMPK) produces a decrease in funny current in HEK cells again as it does in young SA nodal cells, suggesting a low basal AMPK activity. Compound C (inhibition of AMPK) does not have much of an effect on funny current in HEK cells. Therefore, like young nodal cells, this supports the idea that AMPK may not be very active in HEK cells.

Compound C (an inhibitor of AMPK) eliminates the reduction of funny current by AICAR (stimulation of AMPK) in HEK cells.

In the text describing this effect it says "this confirms that the AICAR-induced current reduction is caused by AMPK activation".

I think that this experiment does link the effects of both drugs to AMPK. However, I am not clear why the effect of AICAR is eliminated by Compound C. Is this because inhibition of AMPK by compound C is more effective than stimulation of AMPK by AICAR?

The next piece is a bit confusing to me and I am not sure I am thinking about this correctly. To confirm the activation of AMPK by AICAR, a Western Blot was carried out using specific antibodies to show that there was more phosphorylated versus unphosphorylated AMPK (Fig. S1A). Does AMPK need to be phosphorylated to be active and is this done by AICAR? How does AICAR work?

Has this also been carried out to show that less AMPK was phosphorylated (and hence less active) after exposure to both AICAR and Compound C? Are they both thought to bind to AMPK to modify its activity? Does Compound C limit or reduce phosphorylation of AMPK? If this has not been done, then it might be less easy to rule out the involvement of other proteins.

There is greater current and presumably a greater number of channels at the plasma membrane in HEK cells that have been engineered to express more protein. However, there is also a greater fractional reduction in current by AICAR in the HEK cells. It is stated that

"This supports the view that the action of AMPK activation is not constrained to a restricted pool of channels but rather involves any single channel protein along its pathway from synthesis to membrane expression."

I am not clear on what this means? Is there a relationship between high expression of HCN4 and a proportionally greater effect of AICAR? Also, does this imply that greater internalization of channels from the plasma membrane not an option?

In figure 3, letters for each panel were missing in my version.

Fig. 4. Involvement of hHCN4 serine 1157 in AMPK-mediated channel modulation HEK Cells

Fig. 5. hHCN4 serine 1158 alone does not directly contribute to AMPK-mediated channel modulation. Fig. 6. Loss of AMPK action on double mutant S1157D/S1158D.

These figures showing functional data, along with data showing phosphorylation of S1157 and S1158, are very straight forward and clear to me.

Is heart rate lower in mice that are six months old versus three months old?

Reviewer #2 (Comments to the Authors):

Overall

This manuscript by Palloni et al presents novel data showing that AMPK phosphorylation decreases surface expression of HCN4 channels. Since AMPK is constitutively active in aged hearts and is associated with bradycardia, this mechanism may contribute to the age-related decrease in intrinsic cardiac pacemaker activity. Overall, the studies are sound and findings are important. However, the manuscript's impact would be improved by discussion of how the results relate to previous studies of mechanisms that contribute to the reduction in pacemaker activity with age.

Comments

- The title refers to "age-related bradycardia" and throughout the manuscript there are references to "intrinsic bradycardia" in elderly subjects. However, this phrasing is imprecise. In fact, older individuals do not generally have bradycardia. Rather, the intrinsic pacemaker activity of the heart decreases with age such that maintenance of a normal resting heart rate in older individuals requires a decrease in parasympathetic tone, or even an increase in sympathetic tone, relative to younger individuals. The title and text should be modified accordingly and the concept of intrinsic heart rate should be defined in the introduction.
- The manuscript should discuss the present results in the context of previous studies that found hyperpolarizing shifts in the voltage-dependence of activation of I_f are associated with aging (e.g., Larson et al., 2013). Perhaps differences in experimental protocols contribute to the differences between studies?
- PKA-mediated phosphorylation of S1157 and adjacent residues was previously shown to activate HCN4 channels by depolarizing the voltage-dependence of activation and contributing to the fight-or-flight increase in heart rate (Liao et al, 2010). The manuscript should attempt to reconcile these differences by addressing at least some of the many possible mechanisms by which phosphorylation of S1157 by different kinases could produce different results (e.g., phosphorylation of multiple residues, metabolic context, interaction between the kinases, intracellular compartmentalization, coordinated regulation by phosphorylation and direct cAMP binding, etc etc).
- Pharmacological tools were used at rather high concentrations for fairly long incubation times. Please discuss specificity and possible off-target effects of the AMPK activator AICAR (1 mM for 4 hours) and the AMPK inhibitor, Compound C (30 μ M for 4 hours). Were control cells mock-incubated in vehicle for 4 hours? Information about these experimental procedures should be included in the Methods in addition to the Results.
- It is not how the HEK293F experiments support the conclusion that the AMPK effects are not restricted to a subpopulation of channels but are somehow associated with the biosynthetic pathway of single proteins. Please elaborate.

Reviewer #3 (Comments to the Authors):

Review for Palloni et al., 2025 - AMPK-mediated HCN4 channel phosphorylation contributes to age-related bradycardia.

Here Palloni et al., 2025 show an interesting study that highlights the role of HCN4 channel phosphorylation in age-related bradycardia and that AMP-dependent kinase (AMPK) is key a mediator within this development. They highlighted a key serine residue at position 1157, that after phosphorylation by that AMPK reduces HCN4 membrane expression. Using a mouse model, they highlighted AMPK is constitutively active in an aged population but not within young mice.

The study is well-written, and the experiments presented are well conducted. The hypothesis and approach of this study is logical, and the information generated may be important for understanding the long-term regulation of intrinsic heart rate and ageing mechanisms of Sinus Node Disease. Some considerations to strengthen the manuscript that may wish to be considered are:

- Whilst sex-specific effects were compared (Fig 1) as well as age-related effected (Fig 7) could I be possible that sex-specific effects occur at different ages? I would suggest some clarity around the age of the mice in Fig 1, or further comparison of sex-specific effects in both young and old mice.
- Clarification in text needed around the 4-hour incubation with AICAR, with RT-PCR and FACS it shows a H₂O control. For electrophysiological experiments were any controls recorded after 4-hours incubation in equivalent conditions? If not, the reduced current may be due to the lack of energy availability reducing HCN4 expression (thus current) as mentioned as a mechanism within the discussion.

- Does AICAR have any effect on endogenous currents within HEK296T/F cells as negative control for Figs 3-6.

Reviewer #1 -Point-by-point responses in Italics

In this paper, the action of AMPK on the funny channel of the sinoatrial node is examined. It was shown previously that AMPK activation leads to bradycardia and a reduction of HCN4 at the cell surface of pacemaker cells. Here, it is further shown that the reduction may be linked to direct phosphorylation of specific C-terminal residues of HCN4. Another novel finding is that AMPK may be constitutively active and maintains funny channels at a reduced level in older mice. This could explain the slowing of heart rate in older people in which AMPK may be more active because of changes in metabolism including reduced availability of ATP.

The functional data are very clear, easy to follow and comprehensive, and support the idea that AMPK regulates the surface expression of funny channels. The biochemical data is also very clear and provide support for most of the observations (but not all, see below). I think that a strength of the approach is the ability quantitatively analyze the electrical and biochemical data related to HCN4 and AMPK. However, the cellular and molecular process by which the funny current and channel protein at the plasma membrane is increased is not clear. This is also mentioned in the Discussion and represents limitation of the approach.

-1. Thank you for your review. Our data confirm previous data showing that AMPK activation reduces HCN4/If current, and provides evidence of the molecular mechanism by which this effect is produced, i.e. the phosphorylation of a specific channel protein site. The reviewer is correct in pointing out that this evidence does not clarify exactly which step of the HCN4 trafficking and/or membrane expression is involved. As the reviewer notes we have already mentioned this limitation. A further note stressing this point has now been added to the Discussion (page 26, line last but 2).

Pharmacological activator and inhibitor of AMPK are used which presents both a weakness and a strength. The weakness is that these are used at very high concentrations and therefore it seems at least possible that they are having effects on proteins other than the HCN4 protein and these could be involved in the effects observed. Also, even very specific actions on AMPK may lead to changes in phosphorylation of proteins other than the funny channel. The strength of the use of both drugs is that they produce roughly opposite effects on funny current as would be expected by activation and inhibition of AMPK in both sinoatrial node cells and HEK cells.

Pharmacological activator and inhibitor of AMPK are used which presents both a weakness and a strength. The weakness is that these are used at very high concentrations and therefore it seems at least possible that they are having effects on proteins other than the HCN4 protein and these could be involved in the effects observed.

-2. AICAR is metabolized intracellularly to ZMP (AICAR-PO₄), an AMP analogue (Adamovich et al., 2014). Like AMP, ZMP binds to the regulatory site of the AMPK γ subunit and directly activates AMPK (Kim et al., 2016). Standard AICAR incubation concentrations used in cell cultures for the molecule to exert its action as AMPK activator are in the range of 0.5 to 4 mM; specifically in HEK cells, standard concentrations used are in the range 0.5-1 mM (Adamovich et al., 2014; Thomson et al 2008).

Compound C (dorsomorphin) is an ATP-competitive inhibitor that binds to the conserved catalytic site of the AMPK α subunit, thereby preventing substrate phosphorylation. Structural studies confirm this binding mode (Handa et al., 2011), and functional assays in HEK cells demonstrate inhibition of AMPK activity at a concentration around 30–40 μ M (Handa et al., 2011; Thomson et al., 2008; Zhou et al., 2001).

We have now added a note explaining the mode of action of the drugs and the criteria for choosing drug concentrations in Materials and Methods (Compounds, page 8). The new references quoted have been included in the References list.

Also, even very specific actions on AMPK may lead to changes in phosphorylation of proteins other than the funny channel. The strength of the use of both drugs is that they produce roughly opposite effects on funny current as would be expected by activation and inhibition of AMPK in both sinoatrial node cells and HEK cells.

-3. We have no evidence for the involvement of other AMPK-phosphorylated proteins in the AMPK modulation of HCN4. As the reviewer correctly points out, the demonstration of two opposite effects on HCN4 of the activation/inhibition of the kinase is strong evidence supporting a specific AMPK-dependent control of HCN4 expression. This observation has not been added in Discussion (page 26, line 10).

Another strength of this study is that the effects of these drugs do not alter the current activation curve. The whole-cell measurements are carried out without any cAMP in the pipette. Thus, the increase and decrease in the current is due to an effect specifically on the fully-activated current. In particular, this rules out an effect of current rundown due to a negative shift of the activation curve in the AICAR-induced decrease in current.

A possible weakness is that the effects of the drugs in HCN4-expressing HEK cells may not reflect the process in sinoatrial node cells. However, in HEK cells, a decrease in funny current upon exposure to AICAR (AMPK activator) and the absence of any effect on the current activation curve suggests that AMPK is acting in a similar manner in both cell types. Likewise, lack of effect when AICAR and Compound C are combined in HEK cells follows a similar logic to what is thought to occur in sinoatrial cells. Thus, it seems reasonable to suggest that the regulation of funny channels by AMPK in the sinoatrial pacemaker cells and HEK cells is similar.

-4. We adopted a pharmacological approach in HEK cells exactly because we aimed to reproduce the same changes in AMPK activation as those investigated in SAN myocytes. Having found that the HCN4 current in HEK cells responds to manipulations of AMPK in exactly the same way as I_f in SAN cells supports, as the reviewer points out, identical mechanisms in HEK and SAN cells.

I have some specific comments below.

Fig.1. AMPK activation inhibits I_f in female and male mice pacemaker cells.

These experiments show that funny current is reduced by AMPK activation. Activation of AMPK is carried out by exposing cells to AICAR or control for four hours. The activation curves are not significantly affected by this drug. The current, at fully-activated voltages, is significantly larger after AMPK activation.

The reduction of the funny current by AICAR is clear. However, the concentration of AICAR, at 1mM, seems very high. What is the nature of this drug, how does it work and why is it used at such high concentrations?

-5. See point 2 above for an explanation of the action of the drug and standard incubation concentrations of AICAR.

In later figures, AICAR is shown to not decrease the funny current in older mice. How old were the mice in Figure 1?

-6. *Young mice (3-month). This information has now been included in the legend of Figure 1.*

The use of n=4 female and n=3 male mice is helpful. But it might be easier to follow if a different letter is used for number of animals (n) versus number of cells (also n).

-7. *Nomenclature now modified as requested throughout the text and figure legends, n indicating the number of cells, and N the number of animals.*

I assume that different sets of cells are used to compare the drug with control given the long exposure time (true?).

-8. *The SAN tissue from each mouse was dissociated as described in Materials and Methods. In experiments comparing test vs control cells to analyse the action of a specific drug, we incubated a fraction of the cells (test) in Petri dishes with drug-containing Tyrode solution, and the remaining fraction (control, normally about fifty-fifty) in Petri dishes with vehicle-containing Tyrode. For each experiment, test and control cells were therefore always day-matched and derived from the same mouse. This has been now better clarified in Materials and Methods (page 8, line 12).*

Fig. 7. Age-dependence of AMPK action of I_f in pacemaker cells SAN Cells.

These experiments show that the funny current is decreased by AMPK activation (in young (3 months) but not old (6 months) mice).

Are the mice male or female?

-9. *All the mice used to compare old and young animals were males, as already stated in the first version (now page 22, line 11).*

Fig. 8. AMPK is constitutively activated in old but not in young mice.

These experiments show that the funny current is increased in old (6 months) but not young (3 months) mice by AMPK inhibition (Compound C 30 μ M for 4 hours). Together, these data suggest that AMPK is constitutively activated in older mice.

The increase in funny current of young mice, by Compound C, is also clear. However, even though the concentration is lower than the concentration of AICAR, at 30 μ M it also seems high. What is the nature of Compound C, how does it work and how was this concentration decided upon?

-10. *See answer to point 2 above.*

Are the mice male or female?

-11. *In the first cohort of mice where we tested AICAR incubation we considered separately males and females. Once assessed that in both cases AMPK activation could reduce the I_f current density, we focused our attention on male mice, also based on the observation that the reduction in females was generally lower than in males.*

Female mice naturally experience fluctuations in sex hormone levels; however, we did not investigate if and how their responses vary across different phases of the estrous cycle. Therefore, we concentrated our analysis on male responses.

Fig. 2. AMPK activation (AICAR) reduces membrane expression of HCN4 channels in HEK293 cells.

Fig. 3. Removal of AMPK activation abolishes its action.

AICAR (stimulation of AMPK) produces a decrease in funny current in HEK cells again as it does in young SA nodal cells, suggesting a low basal AMPK activity. Compound C (inhibition of AMPK) does not have much of an effect on funny current in HEK cells. Therefore, like young nodal cells, this supports the idea that AMPK may not be very active in HEK cells.

Compound C (an inhibitor of AMPK) eliminates the reduction of funny current by AICAR (stimulation of AMPK) in HEK cells.

In the text describing this effect it says "this confirms that the AICAR-induced current reduction is caused by AMPK activation".

I think that this experiment does link the effects of both drugs to AMPK. However, I am not clear why the effect of AICAR is eliminated by Compound C. Is this because inhibition of AMPK by compound C is more effective than stimulation of AMPK by AICAR?

-12. As the reviewer notes, this experiment indeed confirms that both drugs act on HCN4 current through AMPK. The data are exactly what expected from the known action of Compound C, an antagonist of ATP that prevents the activation of the AMPK kinase by binding to its catalytic α -subunit; this impairs the AMPK activation induced by AICAR, which instead acts on the regulatory AMPK γ subunit.

The next piece is a bit confusing to me and I am not sure I am thinking about this correctly. To confirm the activation of AMPK by AICAR, a Western Blot was carried out using specific antibodies to show that there was more phosphorylated versus unphosphorylated AMPK (Fig. S1A). Does AMPK need to be phosphorylated to be active and is this done by AICAR? How does AICAR work?

-13. AMPK activation indeed requires its phosphorylation. AMPK is activated around 100-fold when residue Thr-172 of its α 1-subunit is phosphorylated. Like AMP, AICAR binds to the regulatory site 3 of the AMPK γ subunit, leading to the exposure and phosphorylation of the Thr-172 residue and consequent activation of the kinase.

The Western Blot was carried out to show that AICAR does work as an AMPK activator as expected. As mentioned in point 2 above, a paragraph briefly explaining the mode of action of AICAR to activate AMPK has been added to Materials and Methods (Compounds, page 8).

Has this also been carried out to show that less AMPK was phosphorylated (and hence less active) after exposure to both AICAR and Compound C? Are they both thought to bind to AMPK to modify its activity? Does Compound C limit or reduce phosphorylation of AMPK? If this has not been done, then it might be less easy to rule out other the involvement of other proteins.

-14. With AICAR incubation we have shown that there is an increase in the phosphorylation of AMPK, i.e. the active form of AMPK. This confirms well-established data in the literature obtained by Western Blot analysis (see for example Thomson et al., 2008). This same study also demonstrates with Western Blot analysis that Compound C abolishes AICAR-induced AMPK phosphorylation (Thomson et al., 2008, Fig. 4). We have now added this previously published evidence to Results when presenting Figure 3 (page 18, line 3).

There is greater current and presumably a greater number of channels at the plasma membrane in HEK cells that have been engineered to express more protein. However, there is also a greater fractional reduction in current by AICAR in the HEK cells. It is stated that

"This supports the view that the action of AMPK activation is not constrained to a restricted pool of channels but rather involves any single channel protein along its pathway from synthesis to membrane expression."

I am not clear on what this means? Is there a relationship between high expression of HCN4 and a proportionally greater effect of AICAR? Also, does this imply that greater internalization of channels from the plasma membrane not an option?

-15. *The evidence that the inhibitory action of AICAR is not reduced in highly expressing cells suggests that AMPK simply acts on any single HCN4 channel by decreasing, with an equal probability, the chances of a successful membrane expression process. Notice that this is indeed what we find in the subsequent set of experimental data indicating direct phosphorylation of channels. We have now better clarified this consideration in the text explaining Figure 2 and Figure S2 (page 16, line last but 6).*

In figure 3, letters for each panel were missing in my version.

-16. *Thank you, lettering now added*

Fig. 4. Involvement of hHCN4 serine 1157 in AMPK-mediated channel modulation HEK Cells
Fig. 5. hHCN4 serine 1158 alone does not directly contribute to AMPK-mediated channel modulation. Fig. 6. Loss of AMPK action on double mutant S1157D/S1158D.

These figures showing functional data, along with data showing phosphorylation of S1157 and S1158, are very straight forward and clear to me.

-16. *OK, thank you*

Is heart rate lower in mice that are six months old versus three months old?

-17. *We used 3-month (young) and 24-month (old) mice, as mentioned in Results (Figures 7 and 8). Data in the literature report a lower intrinsic heart rate in old compared to young mice (for example 24-month vs 2-3-month mice in Larson et al., 2013, or 19-month vs 4-month in Piantoni et al, 2021). In our work we do not perform in vivo heart rate measurements, but we have reported similar basal resting heart rates (in bmp) for 4-month (568, dark; 515 light) and 19-month mice (576, dark; 529, light) in a previous study (Piantoni et al 2021, Front. Neurosci. 15:617698, Suppl. Material).*

-18. *The following references have been mentioned in the points above and added to the manuscript:*

-Adamovich, Y., Adler, J., Meltser, V., Reuven, N., & Shaul, Y. (2014). AMPK couples p73 with p53 in cell fate decision. *Cell Death and Differentiation*, 21(9), 1451–1459. <https://doi.org/10.1038/cdd.2014.60>

-Handa, N., Takagi, T., Saijo, S., Kishishita, S., Takaya, D., Toyama, M., Terada, T., Shirouzu, M., Suzuki, A., Lee, S., Yamauchi, T., Okada-Iwababu, M., Iwababu, M., Kadowaki, T., Minokoshi, Y., & Yokoyama, S. (2011). Structural basis for compound C inhibition of the human AMP-activated protein kinase $\alpha 2$ subunit kinase domain. *Acta Crystallographica. Section D, Biological Crystallography*, 67(Pt 5), 480–487. <https://doi.org/10.1107/S0907444911010201>

-Kim, J., Yang, G., Kim, Y., Kim, J., & Ha, J. (2016). AMPK activators: Mechanisms of action and physiological activities. *Experimental & Molecular Medicine*, 48(4), e224. <https://doi.org/10.1038/emm.2016.16>

-Thomson, D. M., Herway, S. T., Fillmore, N., Kim, H., Brown, J. D., Barrow, J. R., & Winder, W. W. (2008). AMP-activated protein kinase phosphorylates transcription factors of the CREB family. *Journal of Applied Physiology* (Bethesda, Md.: 1985), 104(2), 429–438. <https://doi.org/10.1152/jappphysiol.00900.2007>

-Zhou, G., Myers, R., Li, Y., Chen, Y., Shen, X., Fenyk-Melody, J., Wu, M., Ventre, J., Doebber, T., Fujii, N., Musi, N., Hirshman, M. F., Goodyear, L. J., & Moller, D. E. (2001). Role of AMP-activated protein kinase in mechanism of metformin action. *The Journal of Clinical Investigation*, 108(8), 1167–1174. <https://doi.org/10.1172/JCI13505>

Reviewer #2 -Point-by-point responses in Italics

Overall

This manuscript by Palloni et al presents novel data showing that AMPK phosphorylation decreases surface expression of HCN4 channels. Since AMPK is constitutively active in aged hearts and is associated with bradycardia, this mechanism may contribute to the age-related decrease in intrinsic cardiac pacemaker activity. Overall, the studies are sound and findings are important. However, the manuscript's impact would be improved by discussion of how the results relate to previous studies of mechanisms that contribute to the reduction in pacemaker activity with age.

1. Thank you for your review. We have now extended the Discussion by comparing our data with previously published data related to mechanisms contributing to age-dependent rate changes, see points 3 and 4 below.

Comments

- The title refers to "age-related bradycardia" and throughout the manuscript there are references to "intrinsic bradycardia" in elderly subjects. However, this phrasing is imprecise. In fact, older individuals do not generally have bradycardia. Rather, the intrinsic pacemaker activity of the heart decreases with age such that maintenance of a normal resting heart rate in older individuals requires a decrease in parasympathetic tone, or even an increase in sympathetic tone, relative to younger individuals. The title and text should be modified accordingly and the concept of intrinsic heart rate should be defined in the introduction.

-2. We are aware of the difference between "intrinsic" and basal resting heart rate and have used the term "age-related bradycardia" in the title only as an approximation. We do agree anyway that the phrasing is not precise and have now modified the title (and chapter heading at page 22). We have also added a sentence in the Introduction mentioning the difference between intrinsic and resting heart rate with specific quotations (Peters et al 2020; Piantoni et al., 2021; Choi et al 2021) (page 4, line 4).

- The manuscript should discuss the present results in the context of previous studies that found hyperpolarizing shifts in the voltage-dependence of activation of I_f are associated with aging (e.g., Larson et al., 2013). Perhaps differences in experimental protocols contribute to the differences between studies?

-3. Although we did not specifically investigate age-related changes of the I_f current properties in control conditions, we did realize that, as the reviewer points out, our datasets in Figures 7 and 8 do not seem to indicate significant age-dependent changes in the position of the activation curve. We do not have an immediate explanation for these differences. We have noticed that in our experiments $V_{1/2}$ values are in the range around -90 mV in both 3-month and 24-month mice, while Larson et al. (2013) data report much more negative values at all ages (about -104, -110 and -117 mV for 2, 24 and 32-month mice, respectively). Some of these differences may originate, as the reviewer suggests, from the different protocols used. We have used a different voltage clamp protocol, as well as different intracellular and extracellular solutions. These considerations and quotations of previous studies have been added to Discussion (page 25, from line last but 4 onwards).

- PKA-mediated phosphorylation of S1157 and adjacent residues was previously shown to activate HCN4 channels by depolarizing the voltage-dependence of activation and contributing to the fight-or-flight increase in heart rate (Liao et al, 2010). The manuscript should attempt to reconcile these differences by addressing at least some of the many possible mechanisms by which phosphorylation of S1157 by different kinases could produce different results (e.g., phosphorylation of multiple

residues, metabolic context, interaction between the kinases, intracellular compartmentalization, coordinated regulation by phosphorylation and direct cAMP binding, etc etc).

-4. Thank you for this comment. Residue S1157 in our study corresponds to S1154 of Liao et al., 2010. Our data demonstrate that the single residue S1157 acts specifically to modulate HCN4 expression, because specific residue ablation totally removes the effect of AMPK phosphorylation. Liao et al (2010) data, on the other hand, indicate the involvement of multiple (4) residues in the PKA-dependent phosphorylating action. This may explain the functional difference in the action of the two kinases.

In Liao et al's (2010) study, inhibition of PKA with a specific inhibitory peptide abolished the β -adrenergic-induced depolarizing shift in the voltage dependence of activation, indicating that phosphorylation within this cluster mediates rapid cAMP-dependent modulation of the channel.

In our work, on the other hand, Ser1157 (corresponding to Ser1154 in Liao et al.) is identified as a specific target for AMPK-dependent phosphorylation, which acts over longer timescales by reducing HCN4 surface expression rather than altering gating properties. Considering that AMPK is activated during energy deprivation, phosphorylation of this site may thus serve as a molecular switch integrating metabolic status with pacemaker activity.

Together these data incidentally lead to the consideration that the region encompassing Thr1153 to Ser1157 may represent a key HCN4 regulatory hotspot.

We have now added these considerations to Discussion (page 24, line last but 7 onwards).

- Pharmacological tools were used at rather high concentrations for fairly long incubation times. Please discuss specificity and possible off-target effects of the AMPK activator AICAR (1 mM for 4 hours) and the AMPK inhibitor, Compound C (30 μ M for 4 hours). Were control cells mock-incubated in vehicle for 4 hours? Information about these experimental procedures should be included in the Methods in addition to the Results.

-5. Reported values of AICAR concentrations necessary for the molecule to exert its action as AMPK activator are in the range 0.5 - 4 mM; specifically in HEK cells, standard concentrations used are in the range 0.5-1 mM (Adamovich et al., 2014). Compound C (dorsomorphin) is an ATP-competitive inhibitor that binds to the conserved catalytic site of the AMPK α subunit, thereby preventing substrate phosphorylation. Structural studies confirm this binding mode (Handa et al., 2011), and functional assays in HEK cells demonstrate inhibition of AMPK activity at concentrations around 30–40 μ M (Handa et al., 2011; Thomson et al., 2008; Zhou et al., 2001).

Yes, control cells were mock-incubated in vehicle for 4 hours.

This has now been indicated more clearly in Materials and Methods (from-page 8, line 12 through to page 9, line 9).

- It is not how the HEK293F experiments support the conclusion that the AMPK effects are not restricted to a subpopulation of channels but are somehow associated with the biosynthetic pathway of single proteins. Please elaborate.

-6. The evidence in Fig. S2 indicates that pharmacological activation of AMPK can modulate the HCN4 current independently of the HCN4 protein expression level. This implies that any single channel protein has the same probability of being phosphorylated independently of the number of channel proteins synthesized. We have now better clarified this consideration (page 16, line last but 6).

-7. The following references have been mentioned in the points above and added to the manuscript:

-Adamovich, Y., Adler, J., Meltser, V., Reuven, N., & Shaul, Y. (2014). AMPK couples p73 with p53 in cell fate decision. Cell Death and Differentiation, 21(9), 1451–1459. <https://doi.org/10.1038/cdd.2014.60>

- Handa, N., Takagi, T., Saijo, S., Kishishita, S., Takaya, D., Toyama, M., Terada, T., Shirouzu, M., Suzuki, A., Lee, S., Yamauchi, T., Okada-Iwabu, M., Iwabu, M., Kadowaki, T., Minokoshi, Y., & Yokoyama, S. (2011). Structural basis for compound C inhibition of the human AMP-activated protein kinase $\alpha 2$ subunit kinase domain. *Acta Crystallographica. Section D, Biological Crystallography*, 67(Pt 5), 480–487. <https://doi.org/10.1107/S0907444911010201>
- Huang X, Yang P, Du Y, Zhang J, Ma A. 2007 Age-related down-regulation of HCN channels in rat sinoatrial node. *Basic Res. Cardiol* 102(5):429–35 [PubMed: 17572839]
- Huang X, Yang P, Yang Z, Zhang H, Ma A. 2016 Age-associated expression of HCN channel isoforms in rat sinoatrial node. *Exp. Biol. Med* 241(3):331–39
- Larson, E. D., St Clair, J. R., Sumner, W. A., Bannister, R. A., & Proenza, C. (2013). Depressed pacemaker activity of sinoatrial node myocytes contributes to the age-dependent decline in maximum heart rate. *Proceedings of the National Academy of Sciences of the United States of America*, 110(44), 18011–18016. <https://doi.org/10.1073/pnas.1308477110>
- Liao, Z., Lockhead, D., Larson, E. D., & Proenza, C. (2010). Phosphorylation and modulation of hyperpolarization-activated HCN4 channels by protein kinase A in the mouse sinoatrial node. *The Journal of General Physiology*, 136(3), 247–258. <https://doi.org/10.1085/jgp.201010488>
- Piantoni C, Carnevali L, Molla D, Barbuti A, DiFrancesco D, Bucchi A, Baruscotti M. (2021) Age-Related Changes in Cardiac Autonomic Modulation and Heart Rate Variability in Mice. *Front Neurosci*. 15:617698. doi: 10.3389/fnins.2021.617698. PMID: 34084126; PMCID: PMC8168539.
- Peters CH, Sharpe EJ, Proenza C. Cardiac Pacemaker Activity and Aging. *Annu Rev Physiol*. 2020 Feb 10;82:21-43. doi: 10.1146/annurev-physiol-021119-034453. Epub 2019 Nov 22. PMID: 31756134; PMCID: PMC7063856.
- Tellez JO, Maczewski M, Yanni J, Sutyagin P, Mackiewicz U, et al. 2011 Ageing-dependent Ca²⁺ remodelling of ion channel and clock genes underlying sino-atrial node pacemaking. *Exp. Physiol* 96(11):1163–78 [PubMed: 21724736] Review of fibrosis in the dysfunction of aging SAN.
- Thomson, D. M., Herway, S. T., Fillmore, N., Kim, H., Brown, J. D., Barrow, J. R., & Winder, W. W. (2008). AMP-activated protein kinase phosphorylates transcription factors of the CREB family. *Journal of Applied Physiology (Bethesda, Md.: 1985)*, 104(2), 429–438. <https://doi.org/10.1152/jappphysiol.00900.2007>
- Zhou, G., Myers, R., Li, Y., Chen, Y., Shen, X., Fenyk-Melody, J., Wu, M., Ventre, J., Doebber, T., Fujii, N., Musi, N., Hirshman, M. F., Goodyear, L. J., & Moller, D. E. (2001). Role of AMP-activated protein kinase in mechanism of metformin action. *The Journal of Clinical Investigation*, 108(8), 1167–1174. <https://doi.org/10.1172/JC113505>

Reviewer #3 -*Point-by-point responses in Italics*

Review for Palloni et al., 2025 - AMPK-mediated HCN4 channel phosphorylation contributes to age-related bradycardia.

Here Palloni et al., 2025 show an interesting study that highlights the role of HCN4 channel phosphorylation in age-related bradycardia and that AMP-dependent kinase (AMPK) is key a mediator within this development. They highlighted a key serine residue at position 1157, that after phosphorylation by that AMPK reduces HCN4 membrane expression. Using a mouse model, they highlighted AMPK is constitutively active in an aged population but not within young mice.

The study is well-written, and the experiments presented are well conducted. The hypothesis and approach of this study is logical, and the information generated may be important for understanding the long-term regulation of intrinsic heart rate and ageing mechanisms of Sinus Node Disease. Some considerations to strengthen the manuscript that may wish to be considered are:

- Whilst sex-specific effects were compared (Fig 1) as well as age-related effected (Fig 7) could I be possible that sex-specific effects occur at different ages?

I would suggest some clarity around the age of the mice in Fig 1, or further comparison of sex-specific effects in both young and old mice.

-1. Thank you for your review. As explained in the Results section of our first version (now at page 22, line 11), we choose to limit our study to male mice based on evidence that 1) the I_f current was slightly larger and 2) the AICAR effect was also moderately stronger. In addition, we wanted to avoid high variability in the action of AMPK activation due to potential fluctuations in hormone levels across the estrous cycle.

Mice in Fig. 1 were 3-month-old. This has now been specified in the figure legend.

- Clarification in text needed around the 4-hour incubation with AICAR, with RT-PCR and FACS it shows a H₂O control. For electrophysiological experiments were any controls recorded after 4-hours incubation in equivalent conditions? If not, the reduced current may be due to the lack of energy availability reducing HCN4 expression (thus current) as mentioned as a mechanism within the discussion.

-2. Yes, in electrophysiological experiments, controls were recorded after 4-hour incubation in equivalent conditions as test cells. The same incubation and the same controls were also used in RT-PCR and FACS experiments. This has now been better clarified in Materials and Methods (page 8 line 12 onwards).

- Does AICAR have any effect on endogenous currents within HEK296T/F cells as negative control for Figs 3-6.

-3. We did run pilot experiments incubating untransfected cells with AICAR as a preliminary control. Untransfected cells do not express HCN4 channels and at -135 mV the recorded currents were very small, in the range 3-4 pA/pF, reflecting lack of significant endogenous components. Mean currents from control and AICAR-treated cells were not significantly different.

Prof. Dario DiFrancesco
University of Milan
Department of Biosciences
via Celoria 26
Milano I-20133
Italy

Re: 202513873R1

Dear Dario,

I am pleased to let you know that your manuscript, titled "AMPK-mediated HCN4 channel phosphorylation contributes to age-related bradycardia", is scientifically acceptable for publication in Journal of General Physiology. Formal acceptance will follow when it is modified in accordance with our editorial policies.

Please note that items needing attention are listed at the bottom of this email under 'manuscript formatting checklist'.

Also, JGP requires a data availability statement for all research article submissions. This statement will be published in the article directly above the Acknowledgments. The statement should address all data underlying the research presented in the manuscript. Please visit the JGP instructions for authors for guidelines and examples of statements at <https://rupress.org/jgp/pages/editorial-policies#data-availability-statement>.

Please submit your final files via this link:
Link Not Available

Thank you for choosing to publish your research in JGP and please feel free to contact me with any questions.

Sincerely,

Jeanne

Jeanne Nerbonne, Ph.D.
On behalf of the Journal of General Physiology

Journal of General Physiology's mission is to publish mechanistic and quantitative molecular and cellular physiology of the highest quality; to provide a best-in-class author experience; and to nurture future generations of independent researchers.

Manuscript formatting checklist:

- MS Word document of text needed (including editable tables)
- MS Word document of supplemental text needed, if applicable (including figure legends and editable tables)
- Brief Statement describing supplementary information needed (in subsection at end of Materials & Methods)
- Please include a data availability statement preceding the Acknowledgments section. Please see <https://rupress.org/jgp/pages/editorial-policies#data-availability-statement>
- FIGURES: Production does not accept JPG formatted figures. Figures created at sufficient resolution and in acceptable format (including supplemental). If working in Illustrator, we prefer .ai or .eps file format. If working in Photoshop please use 600dpi/1000dpi .tiff or .psd file format. Minimum resolution at estimated print size: Minimum resolution for all figures is 600 dpi. For figures that contain both photographs and line art or text, 600 dpi is highly recommended. Figures containing only black and white elements (line art, no color, and no gray) should be 1,000 dpi. Maximum figure size is 7 in wide x 9 in high (17.5 x 22.8 cm) at the correct resolution. <https://jgp.rupress.org/fig-vid-guidelines>
- Supplemental figures, if any, conforming to same guidelines as manuscript figures (noted above)
- If images resemble one from a prior publications, the author must seek permissions (to reproduce or adapt) from the original publisher. [You can resubmit your paper while waiting to hear back from the original publisher but please keep us updated]
- All authors must complete a disclosure form prior to acceptance. A link to complete the form has been sent to all coauthors. Please provide the editorial office with updated email addresses if necessary

Reviewer #1 (Comments to the Authors):

Thank you for your responses to my previous comments.

Reviewer #2 (Comments to the Authors):

The revised manuscript is much improved and I have no further concerns. The authors have done an excellent job clarifying a few issues and reconciling their data with previous literature.

Reviewer #3 (Comments to the Authors):

Thank you for addressing comments.